# ACT-in-LLM: Adaptively Compression Vision Tokens in LLM for High-Resolution Multimodal Large Language Models

## Abstract

High-resolution inputs empower Multimodal Large Language Models (MLLMs) to capture intricate visual details, thereby enhancing comprehension. However, the self-attention mechanism's quadratic complexity poses significant computational and memory challenges as image resolution increases, particularly with long-vision tokens. Existing approaches generally alleviate these issues by reducing vision tokens before feeding them into LLMs. Although efficient, this Pre-LLM compression strategy fails to match the performance of models utilizing all tokens, particularly on high-resolution benchmarks. Our experiments reveal that the performance gap arises from this strategy's limitation in selecting important visual tokens in early LLM layers, leading to the irretrievable loss of critical information. To overcome these challenges, we propose a new strategy that **A**daptively **C**ompresses vision **T**okens with**in** different **LLM** layers, named ACT-in-LLM. Our innovative approach retains all tokens throughout the layers to ensure no vital information is lost while compressing key and value tokens in the self-attention mechanism, to reduce computational costs. The layer-wise compression of ACT-in-LLM is guided by the interaction information between vision and text tokens, leading to more accurate selections. Our theoretical analysis and extensive experiments demonstrate the effectiveness of ACT-in-LLM, showing a 6.3% improvement over existing token compression techniques. It also achieves the competitive performance with non-compression methods, while reducing training/inference time by $\sim 20\%$ and vision tokens by $\sim 60\%$.

## 1 Introduction

In recent years, large language models (LLMs) like GPT-4 (Achiam et al., 2023) and LLaMA (Dubey et al., 2024) have driven advancements in multimodal LLMs (MLLMs), which integrate visual and textual data for better cross-modal understanding (Li et al., 2023c; 2024a; Bai et al., 2023; Zhang et al., 2023; Cheng et al., 2024; Ding et al., 2024). However, MLLMs often process low-resolution visual inputs, limiting fine-grained scene comprehension. While efforts to support high-resolution inputs (Li et al., 2024d; Xu et al., 2024; Li et al., 2024a) exist, they face substantial computational and memory challenges due to the quadratic complexity of self-attention (Vaswani, 2017).

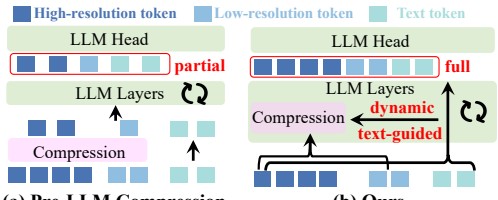

(a) Pre-LLM Compression  (b) Ours

Figure 1: **(a) Pre-LLM Compression Strategy** reduces the number of visual tokens before passing them into the LLM, inevitably leading to information loss. **(b) Our ACT-in-LLM** reserves full tokens for final auto-regressive prediction, while adaptively compressing vision tokens within the specific LLM layers.

To tackle these challenges, existing methods primarily rely on Pre-LLM (Xu et al., 2024; Chen et al., 2024a; Liu et al., 2024c; Huang et al., 2024; Cha et al., 2024) or Early-LLM (Chen et al., 2024a) compression, where the number of visual tokens is reduced before being fed into the LLM or in the early layers of the LLM. This strategy helps lower the computational load and offers competitive

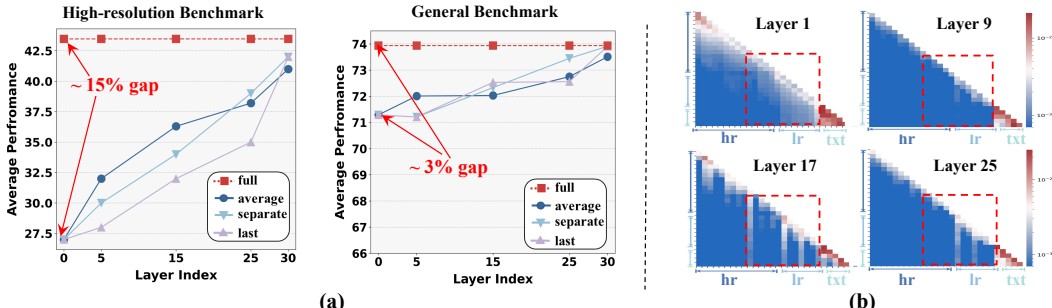

(a)  (b)

Figure 2: **Drawbacks of the early compression. (a) Performance vs. Vision Token Dropping Layers.** The x-axis is the layer index dropping vision tokens, with $0$ indicating before the LLM, and the y-axis is the average performance across benchmarks. We observe that *dropping tokens in earlier layers significantly reduces performance on high-resolution tasks by up to* $15\%$. **(b) Attention Scores Across Layers.** 'hr', 'lr' and 'txt' mean the high-resolution, low-resolution and text tokens respectively. We observe that *early-stage token selection is challenging as low-attention tokens in early layers may gain importance later (see red dotted boxes)*.

performance in general MLLM tasks. However, as shown in Table. 2, there is a notable performance gap ($\sim 9\%$) on high-resolution benchmarks when compared to models that retain all visual tokens.

To investigate this performance gap, we conduct experiments to compare the average performance of compressing vision tokens at different LLM layers on high-resolution and general benchmarks respectively (see Section F.1 for details). We compare four methods: full (all tokens retained), average (dropped based on averaged attention scores), separate (current layer scores, *i.e.*, FastV (Chen et al., 2024a)), and last (final layer scores). In each method, once compression is applied in the specific layer, only $50\%$ of the visual tokens are retained in subsequent layers. Fig.2 (a) shows that the performance gap widens when compression occurs in earlier layers. Additionally, the visualization of the average attention weights of LLaVA-1.5-HD (Fig.2 (b)) shows that vision tokens receiving low attention in early layers become critical in the latter, showing the risks of compressing tokens prematurely. In summary, *existing Pre-LLM approaches compress vision tokens too early, leading to irreversible performance degradation*, potentially due to: (1) early-layer's insufficient interaction between vision and text tokens, (2) varying token importance across layers, making it difficult to decide which tokens to drop, and (3) the inability to recover lost information in latter layers.

To address the above drawbacks, we propose a novel compression strategy, ACT-IN-LLM, which **A**daptively **C**ompresses vision **T**okens **within** different LLM layers. Unlike existing methods that discard tokens prematurely, ACT-IN-LLM retains all tokens across layers, ensuring an implicit error correction mechanism that mitigates the loss of critical information (see Fig. 1 (b)). To reduce computational and memory overhead, ACT-IN-LLM uniformly integrates an adaptive compression module (ACM) into various transformer decoder layers, selectively compressing only the key and value tokens within the self-attention mechanism. Specifically, ACM utilizes the final token in each layer's hidden states— which encodes the complete multimodal context— to guide visual token compression, ensuring more accurate token selection, compared with early-layer selection.

We theoretically demonstrate that this key-value compression used in ACT-IN-LLM provides a better low-rank approximation of the full-token self-attention mechanism compared to the query or all compression used in existing vision token compression techniques. Extensive experiments on high-resolution and general benchmarks, across LLMs of varying sizes (0.5B to 7B parameters), show that ACT-IN-LLM achieves a $6.2\%$ improvement over existing token compression techniques, and competitive performance compared with non-compression models while reducing training/inference time by $\sim 20\%$ and vision tokens by $\sim 60\%$.

## 2 RELATED WORKS

**Multimodal Large Language Models.** Advanced Large Language Models (LLMs) like GPT-4 (OpenAI, 2023), Mistral (Jiang et al., 2023), and Gemini (Team et al., 2023) excel in reasoning, while Multimodal LLMs (MLLMs), such as LLaVA (Liu et al., 2023b), MiniGPT-4 (Zhu et al., 2023), and QwenVL (Bai et al., 2023), extend this to images, though limited resolution hampers fine-grained visual understanding. To address this, splitting images into patches (Bavishi et al.,

2023; Li et al., 2023a) and up-resizing methods (Bai et al., 2023) improve resolution, though they introduce issues like poor visual representation and positional encoding disruptions (Radford et al., 2021). Dual-branch methods add high-resolution branches but increase complexity and training data requirements (Hong et al., 2024; Ding et al., 2023). Cropping strategies offer a more efficient approach by dividing high-resolution images into patches without increasing model parameters (Li et al., 2024d; Xu et al., 2024; Huang et al., 2024). However, increasing image resolution leads to higher computational costs due to the quadratic complexity of self-attention (Vaswani, 2017). This paper is aimed at developing efficient high-resolution MLLMs.

**Vision Token Compression in MLLMs.** Vision Token Compression in MLLMs. Existing methods for vision token compression can be categorized into interaction-based and pre-LLM strategies. Interaction-based approaches (Hong et al., 2024; Li et al., 2024b; Tong et al., 2024) process low-resolution tokens in the LLM while using a high-resolution branch with lightweight cross-attention for feature interaction. However, these methods fail to fully align high-resolution visual inputs with the LLM's low-resolution and text representations, requiring additional parameters and training data (Huang et al., 2024; Yao et al., 2024). Pre-LLM approaches, on the other hand, reduce tokens before entering the LLM, employing either parameterized (Bai et al., 2023; Li et al., 2023c; Cha et al., 2024; Xu et al., 2024) or non-parameterized (Liu et al., 2024a; Yao et al., 2024; Shang et al., 2024) techniques. Recently, Early-LLM methods such as FastV (Chen et al., 2024a) discard tokens during early LLM layers at inference but risk losing critical information due to suboptimal token selection. In contrast, ACT-IN-LLM introduces adaptive token compression at multiple LLM layers, retaining all tokens for final predictions to minimize information loss. Theoretical analysis and experiments demonstrate that ACT-IN-LLM outperforms existing approaches in efficiency and effectiveness.

**Efficient Attention in Transformer Models.** To reduce the computation and memory costs associated with the self-attention mechanism in Transformers, various alternative attention mechanisms have been proposed (Child et al., 2019; Zaheer et al., 2020; Wang et al., 2020; Choromanski et al., 2020). For instance, Sparse Transformer (Child et al., 2019) employs fixed sparse attention patterns to reduce complexity. BIGBIRD (Zaheer et al., 2020) extends this approach by combining multiple attention patterns, including window, random, and global attention, for further efficiency. Reformer (Kitaev et al., 2020) replaces traditional self-attention with locality-sensitive hashing (LSH) to reduce computation costs. Similarly, Axial Transformer (Beltagy et al., 2020) applies attention along single axes of input tensors, significantly lowering the computational burden. PvT-V2 (Wang et al., 2022) leverages the average-pooling to reduce the tokens of the key and value. All of the above methods primarily target single-modal tasks. In contrast, our work focuses on reducing vision tokens based on the multi-modal information in MLLMs for high-resolution multimodal tasks.

## 3 METHOD

### 3.1 OVERALL FRAMEWORK

As illustrated in Fig. 3, ACT-IN-LLM comprises two components: (i) a vision/text tokenizer that processes an image and a question to generate concatenated vision-text embeddings, and (ii) a large language model (LLM) that utilizes these embeddings to predict responses.

**Vision/Text Tokenizer.** The input image is processed using a cropping strategy (Li et al., 2024a;d), producing multiple slices and a low-resolution slice. The original image is resized and padded into a low-resolution slice. To capture fine-grained details, the high-resolution image is dynamically split into slices, with a maximum slice count determined by the base resolution. This allows the image to automatically select an optimal bounding box by calculating the required rows and columns. These slices, along with the low-resolution slice, are then processed through a shared vision tokenizer like CLIP-ViT (Radford et al., 2021), to produce slice-wise vision embeddings. These embeddings are concatenated, with a connector such as a linear layer (Li et al., 2024a) generating the aligned vision representation $\mathbf{H}_0^{\text{vis}} \in \mathbb{R}^{N \times D}$, where $N$ is the total number of vision tokens and $D$ is the embedding dimension. Concurrently, we use the LLM's tokenizer to convert the question into text embeddings, denoted as $\mathbf{H}_0^{\text{txt}} \in \mathbb{R}^{L \times D}$, with $L$ representing the number of text tokens. Finally, the visual and text embeddings are concatenated into $\mathbf{H}_0 = [\mathbf{H}_0^{\text{hr}}, \mathbf{H}_0^{\text{lr}}, \mathbf{H}_0^{\text{txt}}] \in \mathbb{R}^{(N+L) \times D}$ for the LLM.

**Large Language Model (LLM).** Existing LLMs, such as Qwen2 (Yang et al., 2024), LLaMA3 (Dubey et al., 2024) generally consist of several Transformer decoder layers, each of which consists of the multi-head self-attention layer (MSA) and feed-forward network (FFN). The MSA is the critical component of the decoder layer to learn the dense relation between tokens. For-

Figure 3: **Framework of ACT-IN-LLM.** Our ACT-IN-LLM framework follows the general slicing-based MLLMs (Liu et al., 2024b), while applying the adaptive compression module (ACM) at a series of decoder layers of the LLM for computation efficiency, dynamically reducing key/value tokens before the multi-head self-attention (MHA) block, while preserving all query tokens.

mally, given the hidden-states $\mathbf{H}_i \in \mathbb{R}^{(N+L) \times D}$ from the $i$-th layer of the LLM, the single attention head $h \in \{1, 2, ..., H\}$ can be defined as following:

$$\text{head}_{i,h} = \text{Attention}(\mathbf{Q}_{i,h}, \mathbf{K}_{i,h}, \mathbf{V}_{i,h}, \mathbf{M}_i) = \underbrace{\text{softmax}\left(\frac{\mathbf{Q}_{i,h}\mathbf{K}_{i,h}^{\top}}{\sqrt{D}} + \mathbf{M}_i\right)}_{\mathbf{A}_{i,h}} \mathbf{V}_{i,h}, \quad (1)$$

where $\mathbf{Q}_{i,h} = \mathbf{W}_{i,h}^Q \mathbf{H}_i$, $\mathbf{K}_{i,h} = \mathbf{W}_{i,h}^K \mathbf{H}_i$ and $\mathbf{V}_{i,h} = \mathbf{W}_{i,h}^V \mathbf{H}_i$ are the query, key and value matrices, $\mathbf{W}_{i,h}^Q/\mathbf{W}_{i,h}^K \in \mathbb{R}^{D_k \times D}$, $\mathbf{W}_{i,h}^V \in \mathbb{R}^{D_v \times D}$ are the learnable projection matrices, $\mathbf{M}_i$ is the casual mask for the $i$-th layer. $\mathbf{A}_{i,h}$ refers to the attention weight of the the $h$-th head in the $i$-th deocder layer. For clarity, we will not differentiate between $D_k$, $D_v$ and $D$, and just use $D$ in the following. Then the MSA can be represented as:

$$\text{MSA}(\mathbf{Q}, \mathbf{K}, \mathbf{V}) = \text{Concat}(\text{head}_1, \text{head}_2, ..., \text{head}_{i,h})\mathbf{W}_i^O, \quad (2)$$

where $\mathbf{W}_i^O \in \mathbb{R}^{HD_v \times D}$ is the learnable projection matrix, Concat indicates the concatenation operation. The computational complexity for processing all tokens is $O((N + L)^2 \times D)$.

Previous Pre-LLM approaches (Liu et al., 2024b) generally reduce the vision tokens before the LLM (Fig. 1 (a)) leading to several drawbacks as shown in Fig. 2. Differently, our ACT-IN-LLM use the adaptive token compression (ACM) to reduce vision tokens of the key and value within the MSA layer of the LLM, shown as follows:

$$\text{head}_{i,h} = \text{Attention}(\mathbf{Q}_{i,h}, \overline{\mathbf{K}}_{i,h}, \{\overline{\mathbf{V}}_{i,h}), \quad \{\overline{\mathbf{K}}_{i,h}, \overline{\mathbf{V}}_{i,h}\} = \text{ACM}(\{\mathbf{K}_{i,h}, \mathbf{V}_{i,h}\}, \mathbf{A}_{i-1}), \quad (3)$$

where $\mathbf{A}_{i-1}$ is the averaged attention weight from the $i - 1$-th layer, $\{\overline{\mathbf{K}}_{i,h}, \overline{\mathbf{V}}_{i,h}\} \in \mathbb{R}^{(M+L) \times D}$, where $M$ is the number of compressed vision tokens, satisfying $M << N$. In this way, we can reduce the computational complexity of MSA from $O((N + L)^2 \times D)$ of the full tokens to $O((N + L) \times (M + L) \times D)$.

## 3.2 ADAPTIVE COMPRESSION MODULE

As shown in Fig. 2, reducing tokens before the LLM has significant drawbacks, including the absence of text-guided compression, challenges in selecting which tokens to reduce, and the risk of losing important information. To address these issues, our Adaptive Compression Mechanism (ACM) focuses on two key objectives: **(i)** preserving critical vision tokens to prevent information loss, and **(ii)** dynamically compressing tokens based on layer-wise vision-text relations. To achieve the first objective, we retain all query tokens $\mathbf{Q}_i$ across layers,

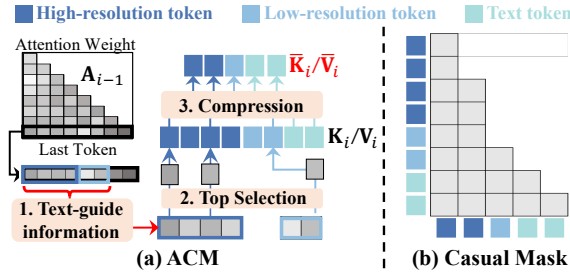

Figure 4: **(a) Adaptive compression module (ACM)** leverages three steps to compress vision tokens of the key and value at the $i$-th layer. **(b) Sampled Casual Mask** for the self-attention layer after ACM.

ensuring an inherent error correction mechanism that mitigates the permanent loss of valuable information. For the second objective, our ACM consists of three steps as shown in Fig. 4 (a).

**Text-guided information extraction.** We utilize attention weights from the previous layer to guide vision token compression, since higher attention weights typically indicate greater importance in the final output (Vaswani, 2017). To leverage the textual information to guide the compression, we focus on the last row of the attention weight, $\mathbf{A}_{i-1}[N + L, :]$, which captures the significance of all previous tokens in relation to the last token. We extract the relevant elements for high-resolution and low-resolution visual tokens as follows:

$$\mathbf{a}_{i-1}^{\text{hr}} = \mathbf{A}_{i-1}[N + L, 0 : N^{\text{hr}}], \quad \mathbf{a}_{i-1}^{\text{lr}} = \mathbf{A}_{i-1}[N + L, N^{\text{hr}} : N^{\text{lr}}], \tag{4}$$

where $N^{\text{hr}}$ and $N^{\text{lr}}$ denote the counts of high-resolution and low-resolution tokens, satisfying $N^{\text{hr}} + N^{\text{lr}} = N$.

**Top Selection.** To retain critical vision tokens, we select the top $N^{\text{hr}}/r_i^{\text{hr}}$ and $N^{\text{lr}}/r_i^{\text{lr}}$ values from $\mathbf{a}_{i-1}^{\text{hr}}$ and $\mathbf{a}_{i-1}^{\text{lr}}$, respectively:

$$\mathbf{s}^{\text{hr}} = \text{Top}(\mathbf{a}_{i-1}^{\text{hr}}, r_i^{\text{hr}}), \quad \mathbf{s}^{\text{lr}} = \text{Top}(\mathbf{a}_{i-1}^{\text{lr}}, r_i^{\text{lr}}), \tag{5}$$

where $\mathbf{s}^{\text{hr}} = \{s_1, s_2, ..., s_{N^{\text{hr}}/r_i^{\text{hr}}}\}$ represents the indices of the top $N^{\text{hr}}/r_i^{\text{hr}}$ values.

**Vision token Compression.** After obtaining the indices, we sampling the $\mathbf{K}_i/\mathbf{V}_i$ and the casual mask $\mathbf{M}_i$ based on $\mathbf{s} = [\mathbf{s}^{\text{hr}}, \mathbf{s}^{\text{lr}}]$, which can be formulated as:

$$\overline{\mathbf{K}}_i = \mathbf{K}_i[\mathbf{s}, :], \quad \overline{\mathbf{V}}_i = \mathbf{V}_i[\mathbf{s}, :], \quad \overline{\mathbf{M}}_i = \mathbf{M}_i[:, \mathbf{s}], \tag{6}$$

where $\overline{\mathbf{K}}_i/\overline{\mathbf{V}}_i \in \mathbb{R}^{(M+L)\times D}, \overline{\mathbf{M}}_i \in \mathbb{R}^{(N+L)\times(M+L)}$ is the sampled casual mask (see the example in Fig. 4 (b)), $M = N^{\text{hr}}/r_i^{\text{hr}} + N^{\text{lr}}/r_i^{\text{lr}}$. Finally, the original self-attention in Eq. 1 can be performed as $\text{Attention}(\mathbf{Q}_{i,h}, \overline{\mathbf{K}}_{i,h}, \overline{\mathbf{V}}_{i,h}, \overline{\mathbf{M}}_i)$.

### 3.3 ARCHITECTURE CONFIGURATIONS

We incorporate the ACM into the decoder layers of the LLM at three stages in a hierarchical way, *i.e.*, $r_i < r_j < r_p$, where $r_i$, $r_j$ and $r_p$ are the sampling ratios in the early, middle and latter layers, based on the observation that the attention weights in the early layers are much dense than the latter ones (see Fig. 2 (b)). Note that for efficiency, we keep the vision tokens index to be identical in each stage. The analysis of different configurations for $r_i/r_j/r_p$ is provided in Table 4a. We uniformly select the $\sim 70\%$ layers among the early, middle and latter layers within LLM decoder layers to apply ACM, for the best performance and efficiency trade-off; see Table 4c and Table 5.

## 4 FORMULATION AND ANALYSIS OF VISION TOKEN COMPRESSION

In this section, we will theoretically show the superiority of our proposed ACM. To this end, we first give a unified formulation of different vision token compression methods in the self-attention mechanism in Section 4.1. Then, we show that ACM is one low-rank approximation of the original self-attention with full tokens under the specific assumption in MLLMs in Section 4.2. Finally, we prove that our ACM provides a better low-rank approximation of the full self-attention mechanism compared to existing vision token compression techniques in Section 4.3.

### 4.1 UNIFIED FORMULATION

For clarity, we omit the layer index $i$ and head index $h$ of Eq. 1 in this section. Formally, the hidden states can be denoted as $\mathbf{H} \in \mathbb{R}^{(N+L)\times D}$, where $N$ and $L$ are the number of the vision tokens (including high-resolution and low-resolution) and the text tokens; see detailed formulation in Section 3.1. Note we omit the system prompt for clarity in our paper. Then, we present a unified and simplified formulation of these different approaches, *i.e.*, presenting the vision compression process as a compression matrix:

$$\mathbf{C} \cdot \mathbf{H} = \mathbf{C} \cdot [\mathbf{V}; \mathbf{T}], \mathbf{C} \in \mathbb{R}^{(M+L)\times(N+L)}, \tag{7}$$

where $\cdot$ is the matrix multiplication, $M$ is the number of the vision tokens after compression, $\mathbf{C}$ is the compression matrix, which is defined as:

$$\mathbf{C} = \begin{pmatrix} \mathbf{C}^{\text{vis}} & \mathbf{O}^1 \\ \mathbf{O}^2 & \mathbf{I} \end{pmatrix}, \tag{8}$$

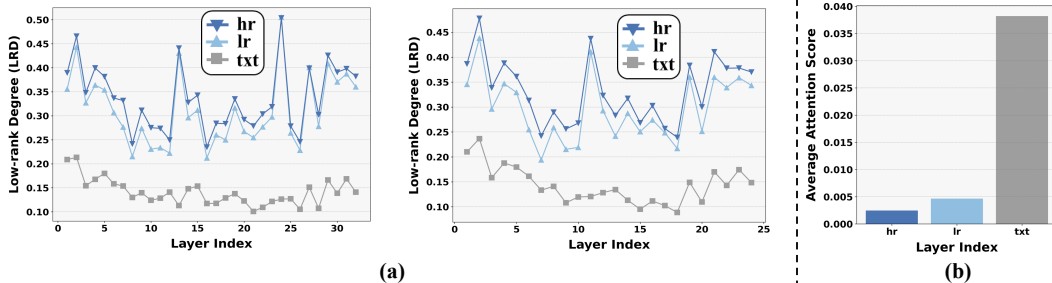

Figure 5: **(a) Low-rank degree.** The x-axis represents the layer index in the LLM and the y-axis is the corresponding LDR (defined in Eq. 15). The high low-rank degree of vision tokens exists in both Vicuna-7b (left) and Qwen2-0.5B (right), and high-resolution tokens show more low-rank than low-resolution ones. **(b) Average attention score.** The x-axis represents the layer index in the LLM and the y-axis is the average attention score. Vision tokens receive small attention on average and high-resolution tokens present less attention than low-resolution ones.

where $\mathbf{C}^{\text{vis}} \in \mathbb{R}^{M \times N}$ is the compression operation for vision tokens, $\mathbf{I} \in \mathbb{R}^{N \times N}$ is the identity matrix, $\mathbf{O}^1 \in \mathbb{R}^{M \times L}$ and $\mathbf{O}^2 \in \mathbb{R}^{L \times N}$ are zero matrices.

Then, the vision token compression in self-attention can be represented as:

$$\text{Com}(\mathbf{C}^Q, \mathbf{C}^K, \mathbf{C}^V) = \text{softmax}\left(\frac{(\mathbf{C}^Q\mathbf{Q})(\mathbf{C}^K\mathbf{K})^\top}{\sqrt{D}}\right) \cdot \mathbf{C}^V\mathbf{V} = \text{softmax}\left(\mathbf{C}^Q\mathbf{A}(\mathbf{C}^K)^\top\right) \cdot \mathbf{C}^V\mathbf{V},$$
(9)

where $\mathbf{C}^Q$, $\mathbf{C}^K$ and $\mathbf{C}^V$ are the compression matrices with the form of Eq. 8 for the query, key and value respectively.

For simplicity, we omit the causal attention mask $\mathbf{M}$ as it does not affect the following analysis. A complete formulation can be found in Eq. 14 of the Appendix.

Using this formulation, *i.e.*, Eq. 9, our ACM can be expressed as $\text{Com}(\mathbf{I}, \mathbf{C}_i^K, \mathbf{C}_i^V)$, where $i$ indicates that performing compression in the $i$-th decoder layer of the LLMs. Similarly, the Pre-LLM compression methods can be represented as $\text{Com}(\mathbf{C}_i^Q, \mathbf{C}_i^K, \mathbf{C}_i^V)$.

### 4.2 LOW-RANK APPROXIMATION

In this section, we would prove that ACM is the low-rank approximation of the self-attention with full tokens based on the formulation of Eq. 9. Note that all proofs of the theorems in this section can be referred to the Appendix.

We first demonstrate that the attention weight of vision tokens, *i.e.*, $\mathbf{A}^{\text{vis}}$, is low-rank:

**Theorem 1.** *For matrix $\mathbf{A}$, and any column vector $\mathbf{v}$ of matrix $\mathbf{V}$, there there exists a matrix $\tilde{\mathbf{A}}$, such that:*

$$\Pr\left(\left\|\tilde{\mathbf{A}}\mathbf{v} - \mathbf{A}\mathbf{v}\right\| \le \epsilon \left\|\mathbf{A}\mathbf{v}\right\|\right) > 1 - o(1) \text{ and } \text{rank}(\tilde{\mathbf{A}}) = \Theta(\log(N)),$$
(10)

where the sub-matrix $\tilde{\mathbf{A}}_{\text{vis}}$ of $\tilde{\mathbf{A}}$ is low-rank.

We also conduct experiments to demonstrate that $\mathbf{A}^{\text{vis}}$ show a higher degree of low-rankness than text tokens in Fig. 5 (a). For more details of this experiment and proof of Eq. 19, refer to the Appendix.

**Assumption 1.** *In the attention weight of MLLMs, vision tokens receive much less attention than text tokens.*

To verify this assumption, Fig. 5 (b) compares the average attention scores of vision and text tokens, showing that text tokens receive significantly more attention ($\sim 13\times$) than vision tokens.

Based on the above theorem and assumption, we show that ACM, *i.e.*, $\text{Com}(\mathbf{I}, \mathbf{C}^K, \mathbf{C}^V)$, can approximate the $\mathbf{A}\mathbf{v}$:

**Theorem 2.** *For the attention weight* $\mathbf{A}$ *and the value* $\mathbf{V}$*, there there exists matrices* $\mathbf{C}^K$ *and* $\mathbf{C}^V$ *in the formulation of Eq.* 8*, such that:*

$$\Pr\left(\left\|\text{softmax}\left(\mathbf{A}(\mathbf{C}^K)^\top\right)\mathbf{C}^V\mathbf{V} - \text{softmax}(\mathbf{A})\mathbf{V}\right\| \le \epsilon \left\|\text{softmax}(\mathbf{A})\mathbf{V}\right\|\right) > 1 - 2e^{-\left(\epsilon^2 - \epsilon^3\right)M/4}. \quad (11)$$

### 4.3 Comparison of different vision token strategies

In this section, we show that our vision token compression strategy *i.e.,* $\text{Com}(\mathbf{I}, \mathbf{C}^K, \mathbf{C}^V)$ is a better approximation of full-token self-attention $\text{Com}(\mathbf{I}, \mathbf{I}, \mathbf{I})$ than existing strategies, such as Pre-LLM or Early-LLM (FastV) (Li et al., 2024a; Chen et al., 2024a) compression $\text{Com}(\mathbf{C}^Q, \mathbf{C}^K, \mathbf{C}^V)$ and FlexAttention (Li et al., 2024b) $\text{Com}(\mathbf{C}^Q, \mathbf{I}, \mathbf{I})$:

| Method | Formulation | Complexity per Layer |
|---|---|---|
| Full Token | $\text{Com}(\mathbf{I}, \mathbf{I}, \mathbf{I})$ | $O((N+L)^2D)$ |
| Pre-LLM/Early-LLM | $\text{Com}(\mathbf{C}_i^Q, \mathbf{C}_i^K, \mathbf{C}_i^V)$ | $O((M+L)^2D)$ |
| FlexAttention | $\text{Com}(\mathbf{C}^Q, \mathbf{I}, \mathbf{I})$ | $O((M+L)(N+L)D)$ |
| Ours | $\text{Com}(\mathbf{I}, \mathbf{C}_i^K, \mathbf{C}_i^V)$ | $O((N+L)(M+L)D)$ |

Table 1: **Comparison of formulation and computation complexity of self-attention operation.** $N$, $M$, $L$ are the number of the original vision tokens, vision tokens after compression and text tokens, respectively. See formulation definition in Eq. 9.

**Theorem 3.** *For any row vector* $\mathbf{a}$ *of* $\mathbf{A}$ *and any column vector* $\mathbf{v}$ *of matrix* $\mathbf{V}$*, any matrices* $\mathbf{C}^Q$*,* $\mathbf{C}^K$ *and* $\mathbf{C}^V$ *in the formulation of Eq.* 8*, if **Theorem 2** holds, then we have:*

$$\Pr(\|\underbrace{\text{Com}(\mathbf{I}, \mathbf{C}^K, \mathbf{C}^V)}_{\text{ACM}} - \underbrace{\text{Com}(\mathbf{I}, \mathbf{I}, \mathbf{I})}_{\text{Full}}\| < \|\underbrace{\text{Com}(\mathbf{C}^Q, \mathbf{C}^K, \mathbf{C}^V)}_{\text{Pre-LLM/Early-LLM}} - \underbrace{\text{Com}(\mathbf{I}, \mathbf{I}, \mathbf{I})}_{\text{Full}}\|) > 1 - o(1) \quad (12)$$

$$\Pr(\|\underbrace{\text{Com}(\mathbf{I}, \mathbf{C}^K, \mathbf{C}^V)}_{\text{ACM}} - \underbrace{\text{Com}(\mathbf{I}, \mathbf{I}, \mathbf{I})}_{\text{Full}}\| < \|\underbrace{\text{Com}(\mathbf{C}^Q, \mathbf{I}, \mathbf{I})}_{\text{FlexAttention}} - \underbrace{\text{Com}(\mathbf{I}, \mathbf{I}, \mathbf{I})}_{\text{Full}}\|) > 1 - o(1) \quad (13)$$

The detailed formulation and complexity comparison with different methods is shown in Table 1.

## 5 Experiments

In this section, we conduct extensive experiments to prove the effectiveness of our proposed ACT-IN-LLM. Specifically, we compare our methods with existing vision token approaches under the same setting in Section 5.1. Then, we show that ACT-IN-LLM can be a plug-and-play method to be applied to different LLMs with different scales in Section 5.2. After that, we demonstrate that our scaling-up ACT-IN-LLM can achieve competitive performance compared with the SOTA MLLMs in section 5.3. Finally, the ablation study of our proposed modules are presented in Section 5.4 for further in-depth analysis.

### 5.1 Comparison with Different Compression methods

**Experiment setting.** To ensure a fair comparison with existing vision token compression methods, we maintain all other settings (*e.g.*, epochs, training dataset, learning rate, cropping strategies, number of slices from high-resolution images, etc.) constant, varying only the method of vision token compression to compare their respective performances. Specifically, we utilize CLIP-ViT-L/14-224px as the vision encoder and Vicuna-7B-v1.5 as the LLM. We adopt a two-stage training approach comprising a pre-training stage and an instruction supervised (SFT) fine-tuning stage, following the training parameters outlined in (Liu et al., 2023a). The number of slices is set to four, consistent with LLaVA-1.5-HD (Liu et al., 2023a). All methods ultimately compress visual tokens of high-resolution slices to $\sim 256$ for fairness.

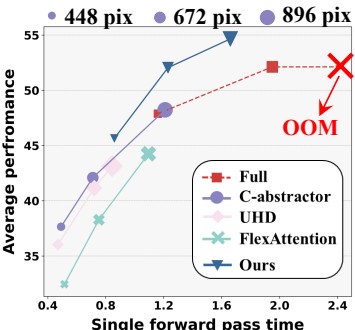

Figure 6: **Trade-off of different methods.** 'OOM' indicates the out-of-the-memory.

**Training dataset.** For pre-training, we follow (Liu et al., 2023b) and use a $558K$ subset of the LAION-CC-SBU dataset with BLIP captions (Li et al., 2023c). For supervised fine-tuning, in addition to the original $665K$ data from LLaVA, we gather

| Model | Efficiency | | General | | | High-Resolution | | | | |
|---|---|---|---|---|---|---|---|---|---|---|
| | Times(ms) | Memory(GB) | SEED | POPE | MME | VQA-text | ChartQA | DocVQA | InfoVQA | Average |
| Full | 621(**100.0%**) | 19.9(**100.0%**) | 64.2 | 85.3 | 1466.7 | 60.5 | 49.0 | 46.5 | 35.0 | 48.0 |
| Q-former | 507(**81.6%**) | 18.6(**93.0%**) | 61.4 | 84.2 | 1432.5 | 53.6 | 21.8 | 21.6 | 25.4 | 30.6 |
| Avg-pooling | 461(**74.3%**) | 18.1(**90.7%**) | 61.5 | 85.2 | 1402.7 | 56.5 | 37.8 | 34.9 | 27.4 | 39.1 |
| FlexAttn | 505(**81.3%**) | 18.6(**93.4%**) | 60.0 | 87.3 | 1442.9 | 53.6 | 27.3 | 24.3 | 24.6 | 32.5 |
| LLaVA-UHD | 470(**75.7%**) | 18.3(**91.9%**) | 60.5 | 85.8 | 1407.5 | 54.2 | 33.2 | 29.9 | 26.9 | 36.0 |
| C-Abstractor | 492(**79.2%**) | 18.2(**91.6%**) | 62.1 | 86.6 | 1448.1 | 56.7 | 36.4 | 31.3 | 26.2 | 37.6 |
| FastV | 499(**80.3%**) | 18.3(**94.0%**) | 61.9 | 86.7 | 1412.9 | 58.1 | 35.0 | 38.6 | 27.7 | 39.9 ↘ |
| Ours | 515(**83.0%**) | 18.8(**94.0%**) | **63.5** | **87.6** | **1480.3** | **58.5** | **46.1** | **45.2** | **31.6** | **45.4** +5.5 |
| FastV w/o train | 499(**80.3%**) | 18.3(**94.0%**) | 61.5 | 85.8 | 1412.9 | 57.8 | 33.5 | 37.3 | 26.2 | 38.7 |
| Ours w/o train | 515(**83.0%**) | 18.8(**94.0%**) | 63.2 | 87.1 | 1443.2 | 58.3 | 43.2 | 42.8 | 29.8 | 43.5 |

Table 2: **Comparison with SOTA vision token compression methods.** The ratios of time and memory cost for different methods relative to the full method are highlighted in **(green)**. All models are trained in the same setting. Gray means the model without vision token compression. 'w/o train' means the direct using our method without training. **Bold** means the best value and Underline mean the second-best value. The number in blue indicates the difference to the prior state of the art.

additional public datasets from high-resolution benchmarks, including ChartQA (Masry et al., 2022), DocVQA (Mathew et al., 2021), and InfoVQA (Mathew et al., 2022), yielding a total of $774K$ data.

**Evaluation dataset.** We evaluate different methods on both high-resolution benchmarks including VQA-text(Singh et al., 2019), ChartQA val set (Masry et al., 2022), DocVQA val set (Mathew et al., 2021), InfoVQA val set (Mathew et al., 2022), and general multimodal benchmarks including SEED (Li et al., 2023b), POPE (Li et al., 2023d), MME (Fu et al., 2023).

**Results.** We compare our method with state-of-the-art pre-LLM approaches (e.g., Q-former (Li et al., 2023c), Avg-pooling (Li et al., 2024a), LLaVA-UHD (Xu et al., 2024), and C-Abstractor (Cha et al., 2024)) and interaction approaches (e.g., FlexAttention (Li et al., 2024b)), as well as FastV (Chen et al., 2024a). From Table 2, our method demonstrates a superior trade-off compared to existing approaches, *e.g.*, achieving $82.96\%$ of the single-forward time of the full tokens while attaining $45.35\%$ average performance on high-resolution benchmarks, outperforms $5.5\%$ over the previous SOTA. Without training, our method can also outperform other vision token compression approaches even if they are trained.

Furthermore, we analyze the trade-offs of various vision token compression approaches by reporting average performance on high-resolution benchmarks alongside the single-example forward pass time at different input resolutions, executed on one V100 GPU. As shown in Fig. 6, our method demonstrates a superior trade-off, particularly as image resolution increases, indicating its effectiveness in balancing performance and efficiency, around $65\%$ times compared with the full model while achieving the competitive performance.

## 5.2 SCALING UP ACT-IN-LLM

**Experiment setting.** This section investigates whether our ACT-IN-LLM performance improves with increasing model size and SFT dataset size. We employ the pre-trained InternViT-300M (Chen et al., 2024b) as our vision encoder, evaluating various scale LLMs (Qwen2-0.5B (Yang et al., 2024), Phi3-3B (Abdin et al., 2024), and IntermLM2-7B (Cai et al., 2024)) alongside SFT data sizes of $0.5M$, $0.7M$, and $1.2M$. Average performance on high-resolution benchmarks is reported in Fig. 7.

**Impact of LLM Scale.** As illustrated in Fig.7, our ACT-IN-LLM shows consistent improvement with increasing model size across different SFT data scales. For example, with $0.7M$ SFT data, ACT-IN-LLM(0.5B) achieves an average score of $54.58\%$, while ACT-IN-LLM(3B) reaches $67.00\%$, resulting in a $6.23\%$ gain when scaling from 3B to 7B.

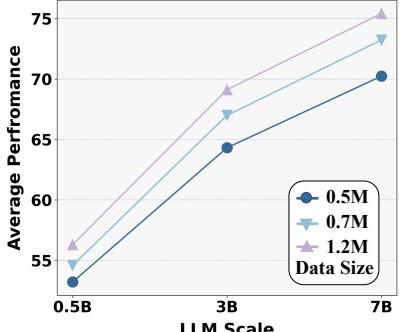

Figure 7: **Effect of different model and data sizes.**

**Impact of SFT Data Size.** Fig.7 also indicates that training with larger SFT datasets enhances ACT-IN-LLM performance across various LLM sizes. Specifically, increasing from $0.5M$ to $0.7M$

| Model | Max Tokens$^V$ | Data size | General | | | | High-Resolution | | | | |
|---|---|---|---|---|---|---|---|---|---|---|---|
| | | | SEED | GAQ | POPE | MME | VQA-text | ChartQA$^{test}$ | DocVQA$^{test}$ | InfoVQA$^{test}$ | AVG |
| *Without Vision Token Compression* | | | | | | | | | | | |
| LLaVA-Next | 2880 | 760K | 72.7 | **65.2** | - | 1519 | 64.9 | 69.5 | 72.6 | - | - |
| Mini-Gemini-HD | 2880 | 1.5M | **73.2** | 64.5* | 86.0* | 1546 | **68.4** | 53.5* | 56.1* | 39.5* | 53.4* |
| LLaVA-Onevision | 7290 | 4.8M | - | - | - | 1580 | - | 80.0 | 90.2 | 70.7 | - |
| InternVL2 | 3072 | >5M | 70.7 | 63.2 | **86.9** | **1648** | **77.4** | **83.3** | **91.6** | **74.8** | **81.8** |
| *With Vision Token Compression or $\leq$ 1k tokens* | | | | | | | | | | | |
| LLaVA-FlexAttn | ~576 | 665K | 62.8* | 62.2 | 85.9 | **1479** | 48.9 | - | - | - | - |
| UReader | ~841 | - | - | - | - | - | 57.6 | 59.3 | 65.4 | 42.2 | 56.1 |
| TextMonkey | 768 | 409K | - | - | - | - | 65.9 | 65.5 | 71.5 | 28.2 | 57.8 |
| DOCOWL2 | 324 | 6.4M | - | - | - | - | 66.7 | 70.0 | 80.7 | 46.4 | 65.9 |
| Cambrian-1 | 576 | 10M | - | - | - | - | **71.7** | 73.3 | 77.8 | - | - |
| ACT-IN-LLM (Ours) | ~1K | 1.2M | **71.3** | **64.4** | **86.1** | 1523 | 71.4 | **77.3** | **81.0** | **55.9** | **71.4** |

Table 3: **Comparison with the stat-of-the-art MLLMs.** 'Tokens$^V$' means the vision token numbers. The LLM size of all models is around 7B. Within each group, the best and the second-best values are marked in **Bold** and Underline. * means the results reproduced by ours using official checkpoints.

yields approximately $2\%$ improvement, while moving to $1.2M$ data further boosts performance by around $6\%$ relative to $0.7M$ data.

In summary, as both model size and SFT data increase, our method consistently achieves significant gains, indicating its potential applicability for training larger-scale models and datasets.

## 5.3 STATE-OF-THE-ART COMPARISON

Our ACT-IN-LLM utilizes the 7B LLM trained on $1.2M$ SFT data, achieving the best performance as detailed in Section 5.2. We compare our model against state-of-the-art (SOTA) MLLMs, including: (i) MLLMs without vision token compression: LLaVA-NeXT (Liu et al., 2024b), Mini-Gemini-HD (Li et al., 2024c), LLaVA-Onevision (Li et al., 2024a), and InternVL2 (Chen et al., 2024b) and (ii) MLLMs with vision token compression or the vision tokens $\leq$ 1K: LLaVA-FlexAttn (Li et al., 2024b), UReader (Ye et al., 2023), Cambrian-1 (Tong et al., 2024), TextMonkey (Liu et al., 2024c) and DOCOWL2 (Hu et al., 2024).

Table 3 summarizes the performance of different methods alongside the maximum number of vision tokens and STF data sizes. From the table, we can observe that our ACT-IN-LLM obtains the SOTA performance on both general and high-resolution benchmarks among the MLLMs in the second group. Even compared with the MLLMs in the first group those using exceed 3K tokens, our ACT-IN-LLM achieves $87.2\%$ of InternVL2's performance on high-resolution benchmarks while utilizing only $32.8\%$ of the vision tokens and less than $24\%$ of the SFT data, highlighting its efficiency.

## 5.4 ABLATION STUDY

In this section, we evaluate the effectiveness of the Adaptive Compression Module (ACM), a pivotal component of ACT-IN-LLM. Using the baseline configuration outlined in Section 5.1, our ablation study addresses three critical aspects: *(i) compression ratios—quantifying vision token reduction, (ii) compression implementation methods—strategies for token compression, and (iii) compression layers—optimal layers for token reduction.*

**Compression Ratios.** Compression ratios dictate the number of vision tokens reduced, specifically characterized by $r_i^{hr}$ and $r_i^{lr}$, which represent the reduction ratios for high-resolution and low-resolution tokens at layer $i$. We categorize the LLM layers into three types: the early layers with compression ratio of $r_i^{hr}/r_i^{hr}$, the middle with compression ratio of $r_j^{hr}/r_j^{hr}$ and the latter layers with compression ratio of $r_k^{hr}/r_k^{hr}$. To explore the best configuration of the compression ratios, our ablation study consists of two steps. First, we maintain equal compression ratios for he high-resolution and low-resolution vision tokens ($r^{hr} = r^{lr}$) and change $r$ across different layer groups, *i.e.*, plain type ($r_i = r_j = r_p$) and hierarchical type ($r_i \neq r_j \neq r_p$). Results (rows (a) to (e) in Table 4a) indicate that a hierarchical approach ($r_i < r_j < r_p$) outperforms the plain type, aligning with the observed trend of sparser attention in deeper layers (Fig.2(b)). Subsequently, we investigate distinct ratios for high- and low-resolution tokens within each layer. The results from row (f) to row (i) in Table 4a demonstrate that $r^{hr} > r^{lr}$ performs better than $r^{hr} \leq r^{lr}$, likely due to the higher low-rank nature of high-resolution tokens (Fig. 5).

| $\{r_l^{hr}/r_l^{lr}\}_{l=i,j,p}$ | *time* | *general* | *hr* |
|---|---|---|---|
| *plain*: $r_i = r_j = r_p$ | | *identical*: $r^{hr} = r^{lr}$ | |
| (a) $\{2/2, 2/2, 2/2\}$ | 563 | 74.25 | **45.89** |
| (b) $\{4/4, 4/4, 4/4\}$ | 516 | 74.01 | 44.18 |
| (c) $\{8/8, 8/8, 8/8\}$ | **506** | 74.12 | 43.25 |
| *hierarchical*: $r_i \neq r_j \neq r_p$ | | *identical*: $r^{hr} = r^{lr}$ | |
| (d) $\{2/2, 4/4, 8/8\}$ | 513 | 74.86 | 44.95 |
| (e) $\{8/8, 4/4, 2/2\}$ | 513 | 74.42 | 43.35 |
| *hierarchical*: $r_i \neq r_j \neq r_p$ | | *distinct*: $r^{hr} \neq r^{lr}$ | |
| (f) $\{2/1, 4/1, 8/1\}$ | 531 | 75.02 | 45.11 |
| (g) $\{2/2, 4/2, 8/2\}$ | 515 | 74.98 | 45.12 |
| (h) $\{2/1, 4/2, 8/4\}$ | 515 | **75.04** | 45.35 |
| (i) $\{2/4, 4/4, 8/4\}$ | 513 | 74.23 | 44.51 |

(a) **Compression ratios** $r$. The detailed definition of $\{r_i^{hr}/r_i^{lr}, r_j^{hr}/r_j^{lr}, r_p^{hr}/r_p^{lr}\}$ is presented in Section 3.2.

| way | *general* | *hr* |
|---|---|---|
| Attention-weight | 75.04 | **45.35** |
| AvgPool-1D | **75.06** | 45.08 |
| AvgPool-2D | 74.12 | 43.56 |
| Learnable Projection | 74.07 | 42.21 |
| Pre-LLM | 72.28 | 39.15 |

(b) **Compression ways**. 'Pre-LLM': the best Pre-LLM approaches in Table 2.

| layer postions | *general* | *hr* |
|---|---|---|
| early | 73.51 | 42.33 |
| middle | 74.68 | 44.08 |
| latter | 7430 | 44.29 |
| uniform | **75.04** | **45.35** |

(c) **Compression layer positions**. There are totally 20 layers to compress tokens for fair compression.

Table 4: **ACM module ablation experiments**. **time**: single-forward pass time (ms). **general**: the average performance on general benchmarks. **hr**: the average performance on high-resolution benchmarks. Best results and default settings are reported in **Bold** and ⬛ gray ⬛.

**Compression Ways.** In Section 3.2, we use the attention weight $\mathbf{A}_{i-1}$ from the $i-1$-th layer to guide the vision token compression in the $i$-th layer. We compare the attention weight with three alternative compression methods, including (i) average-pooling 1D: directly apply average-pooling 1D to the vision tokens of $\mathbf{K}_i/\mathbf{V}_i$. (ii) average-pooling 2D: reshape the vision-tokens in $\mathbf{K}_i/\mathbf{V}_i$ to be 2D, and then apply average-pooling 2D to the reshaped 2D hidden states. (iii) Learnable projection: use a learnable projection to reduce the length of $\mathbf{K}_i/\mathbf{V}_i$.

Table 4b reports the average performance[1] of general and high-resolution benchmarks of different implementations. Results show that all different ways of ACM can outperform Pre-LLM approaches, confirming the effectiveness of our method. Non-parameter ways consistently yield better performance than parameterized ones (*e.g.*, learnable projection), possibly due to the learnable methods requiring additional training complexity of learning effective mappings to compress vision tokens.

**Compression Layers.** We compress vision tokens in a uniform way as described in Section 3.3, *i.e.*, uniformly reducing tokens in early, middle and latter layers. To ablate where to reduce vision tokens, we further compare the other three types of compression layers: (i) the first 70% layers, (ii) the middle 70% layers and (iii) the last 70% layers. From Table 4c, we can find that selecting 70% layers uniformly across all layers of the LLM to apply ACM achieves the best performance. We also experiment with different proportions of ACM layers

| num | **times** | **general** | **high-resolution** |
|---|---|---|---|
| 50% | 552 | **75.25** | **46.12** |
| 60% | 538 | 75.02 | 45.68 |
| 70% | 515 | 75.04 | 45.35 |
| 80% | 512 | 74.19 | 44.61 |
| 90% | **502** | 74.08 | 43.17 |

Table 5: **Effect of the ratio of ACM layers among all layers.** Best results and default settings are reported in **Bold** and ⬛ gray ⬛.

applied across all LLM layers in Table 5, showing that using more ACM layers would improve the efficiency while degrading the performance. The detailed layer indexes of incorporating ACM can be found in the Appendix.

## 6 CONCLUSION

In this paper, we introduce ACT-IN-LLM, which enhances the efficiency of multimodal large language models (MLLMs) by adaptively compressing vision tokens across different LLM layers. Unlike prior methods that reduce vision tokens before LLM processing, our approach retains all tokens, providing an inherent error correction mechanism to prevent the loss of critical information. Additionally, the layer-wise compression is guided by interactions between vision and text tokens, ensuring precise token selection. Our theoretical analysis and extensive experiments demonstrate that ACT-IN-LLM outperforms existing vision token compression techniques. Moreover, we reveal the potential for scaling up ACT-IN-LLM to achieve competitive performance even with SOTA MLLMs without vision token compression.

---

[1]The MME Perception score is scaled down by 20 to align with other datasets, as (Tong et al., 2024).

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

## A   FORMULATION OF SELF-ATTENTION

We can transfer Eq. 1 to the following formulation:

$$\text{Self-Attn}(\mathbf{Q}, \mathbf{K}, \mathbf{V}) = \text{softmax}\left(\frac{(\mathbf{Q} \odot \mathbf{M})(\mathbf{K^T} \odot \mathbf{M})}{\sqrt{\overline{D}}}\right)\mathbf{V}, \tag{14}$$

where $\odot$ indicates the element-wise multiplication. Here we omit the layer and head indexes.

## B   LOW-RANK ANALYSIS

**Experiment Setting.** We use a mainstream high-resolution MLLM, the LLaVA-1.6-7B with $2 \times 2$ high-resolution slices (), as our baseline model. Then, we randomly sample 50 samples from three common high-resolution benchmarks, *i.e.*, ChartQA, DocVQA and InfoVQA, as input, and obtain the average attention weight $\{\mathbf{A}_i\}_{i=1}^{32}$ across different samples, where $\mathbf{A}_i$ is the attention weight of the $i$-th LLM layer in LLaVA-1.6-7B. For better analysis of different types of tokens, we divide the attention weight $\mathbf{A}$ into three different sub-matrices : $\mathbf{A}^{\text{hr}} \in \mathbb{R}^{N^{\text{hr}} \times N^{\text{hr}}}$, $\mathbf{A}^{\text{lr}} \in \mathbb{R}^{N^{\text{lr}} \times N^{\text{lr}}}$ and $\mathbf{A}^{\text{txt}} \in \mathbb{R}^{N^{\text{txt}} \times N^{\text{txt}}}$.

**Low-Rank Degree Measurement.** In high-resolution MLLMs, the disparity in the number of different types of tokens is significant., *e.g.*, $N^{\text{hr}} >> N^{\text{lr}}/N^{\text{txt}}$. We introduce LRD to better measure the low-rank degrees for matrices with different sizes. Formally, we can conduct the singular value decomposition for one attention weight matrix $\mathbf{A}$, obtain its corresponding singular values $R$. Then, the low-rank degree can be computed as following:

$$\text{LRD} = \text{sum}(\text{Top}(r * N^{\text{sv}}, R))/\text{sum}(R), \tag{15}$$

where $\text{Top}(a, b)$ indicates selecting the top $a$ singular values from $b$, $r \in (0, 1)$ is the ratio of sampling singular values and $N^{\text{sv}}$ is the number of singular values in $R$. In this way, LDR reflects the proportion of the sum of the top $r\%$ singular values to the total sum of all singular values. A larger LDR indicates a higher degree of low-rankness in the matrix, and vice versa.

## C   PROOF OF THEOREM 1

*Proof.* We first write $\mathbf{A}$ and $\mathbf{v}$ as:

$$\mathbf{A} = \begin{pmatrix} \mathbf{A}_{\text{vis}} & \mathbf{A}_1 \\ \mathbf{A}_2 & \mathbf{A}_3 \end{pmatrix}, \quad \mathbf{v} = (\mathbf{v}_{\text{vis}} \quad \mathbf{v}_1) \tag{16}$$

Then, $\mathbf{Av}$ can be represented as:

$$\begin{pmatrix} \mathbf{A}_{\text{vis}}\mathbf{v}_{\text{vis}} + \mathbf{A}_1\mathbf{v}_1 \\ \mathbf{A}_2\mathbf{v}_{\text{vis}} + \mathbf{A}_3\mathbf{v}_1 \end{pmatrix} \tag{17}$$

Similarly, $\tilde{\mathbf{A}}$ can be formulated as:

$$\begin{pmatrix} \tilde{\mathbf{A}}_{\text{vis}}\mathbf{v}_{\text{vis}} + \tilde{\mathbf{A}}_1\mathbf{v}_1 \\ \tilde{\mathbf{A}}_2\tilde{\mathbf{v}}_{\text{vis}} + \tilde{\mathbf{A}}_3\mathbf{v}_1 \end{pmatrix} \tag{18}$$

We can let $\mathbf{A}_1\mathbf{v}_1 = \tilde{\mathbf{A}}_1\mathbf{v}_1$, $\mathbf{A}_2\mathbf{v}_{\text{vis}} = \tilde{\mathbf{A}}_2\tilde{\mathbf{v}}_{\text{vis}}$ and $\mathbf{A}_3\mathbf{v}_1 = \tilde{\mathbf{A}}_3\mathbf{v}_1$, then the proof of Eq. 10 can be equal to prove that there exists a low-rank matrix $\tilde{\mathbf{A}}_{\text{vis}}$ such that:

$$\Pr\left(\left\|\tilde{\mathbf{A}}_{\text{vis}}\mathbf{v}_{\text{vis}} - \mathbf{Av}\right\| \leq \epsilon \|\mathbf{Av}\|\right) > 1 - o(1) \text{ and } \text{rank}(\tilde{\mathbf{A}}) = \Theta(\log(N)). \tag{19}$$

The main idea of the proof follows (Lindenstrauss & Johnson, 1984; Arriaga & Vempala, 2006; Wang et al., 2020). Based on the distributional Johnson–Lindenstrauss lemma (Arriaga & Vempala, 2006), for any $\mathbf{x} \in \mathbb{R}^{1 \times N}, \mathbf{y} \in \mathbb{R}^{N \times 1}$, we have:

$$\Pr\left(\|\mathbf{x}\mathbf{R}^\top\mathbf{Ry} - \mathbf{xy}\| \leq \epsilon\|\mathbf{xy}\|\right) > 1 - 2e^{-\left(\epsilon^2 - \epsilon^3\right)M/4}, \tag{20}$$

where $\mathbf{R} \in \mathbb{R}^{N \times M}$. After constructing a low-rank matrix $\tilde{\mathbf{A}}_{\text{vis}} = \mathbf{A}_{\text{vis}} \mathbf{R}^\top \mathbf{R}$, for any row vector $\mathbf{a} \in \mathbf{A}_{\text{vis}}$ and any column vector $\mathbf{v} \in \mathbf{V}$, we have:

$$\Pr\left(\left\|\mathbf{a}\mathbf{R}^\top \mathbf{R}\mathbf{v} - \mathbf{a}\mathbf{v}\right\| \leq \epsilon \|\mathbf{a}\mathbf{v}\|\right) > 1 - 2e^{-\left(\epsilon^2 - \epsilon^3\right)M/4}. \tag{21}$$

Therefore, we have:

$$
\begin{aligned}
\Pr\left(\left\|\tilde{\mathbf{A}}_{\text{vis}}\mathbf{v}_{\text{vis}} - \mathbf{A}\mathbf{v}\right\| \leq \epsilon \|\mathbf{A}\mathbf{v}\|\right) &= \Pr\left(\left\|\mathbf{A}_{\text{vis}}\mathbf{R}^\top \mathbf{R}\mathbf{v}_{\text{vis}} - \mathbf{A}\mathbf{v}\right\| \leq \epsilon \|\mathbf{A}\mathbf{v}\|\right) \\
&\geq 1 - \sum_{\mathbf{a} \in P} \Pr\left(\left\|\mathbf{a}\mathbf{R}^\top \mathbf{R}\mathbf{v} - \mathbf{a}\mathbf{v}\right\| > \epsilon \|\mathbf{a}\mathbf{v}\|\right) \\
&> 1 - 2Ne^{-\left(\epsilon^2 - \epsilon^3\right)M/4} = 1 - o(1)
\end{aligned}
\tag{22}
$$

## D PROOF OF THEOREM 2

*Proof.* Based on the definition of $(\mathbf{C}^K)^\top / \mathbf{C}^V$ (see Eq. 8), $\text{softmax}\left(\mathbf{a}((\mathbf{C}^K)^\top)\right) \mathbf{C}^V \mathbf{v}$ can be represented as:

$$\text{softmax}\left(\mathbf{a}_{\text{vis}}(\mathbf{C}_{\text{vis}}^K)^\top, \quad \mathbf{a}_1\right) \begin{pmatrix} \mathbf{C}_{\text{vis}}^V \mathbf{v}_{\text{vis}} \\ \mathbf{v}_1 \end{pmatrix}. \tag{23}$$

Similarly, $\text{softmax}(\mathbf{a})\mathbf{v}$ can be presented as:

$$\text{softmax}\left(\mathbf{a}_{\text{vis}}, \quad \mathbf{a}_1\right) \begin{pmatrix} \mathbf{v}_{\text{vis}} \\ \mathbf{v}_1 \end{pmatrix}. \tag{24}$$

Let define:

$$D_1 = \text{sum}(\exp(\mathbf{a}_{\text{vis}}(\mathbf{C}_{\text{vis}}^K)^\top) + \exp(\mathbf{a}_1)) \tag{25}$$

$$D_2 = \text{sum}(\exp(\mathbf{a}_{\text{vis}}) + \exp(\mathbf{a}_1)) \tag{26}$$

Then, Eq. 23 and Eq. 24 can be formulated as:

$$\exp(\mathbf{a}_{\text{vis}}(\mathbf{C}_{\text{vis}}^K)^\top)\mathbf{C}_{\text{vis}}^V \mathbf{v}_{\text{vis}}/D_1 + \exp(\mathbf{a})\mathbf{v}_1/D_1. \tag{27}$$

$$\exp(\mathbf{a}_{\text{vis}})\mathbf{v}_{\text{vis}}/D_2 + \exp(\mathbf{a})\mathbf{v}_1/D_2. \tag{28}$$

Then, we have:

$$
\begin{aligned}
\left\|\text{softmax}\left(\mathbf{a}(\mathbf{C}^K)^\top\right) \mathbf{C}^V \mathbf{v} - \text{softmax}(\mathbf{a})\mathbf{v}\right\| &= \|\exp(\mathbf{a}_{\text{vis}}(\mathbf{C}_{\text{vis}}^K)^\top)\mathbf{C}_{\text{vis}}^V \mathbf{v}_{\text{vis}}/D_1 - \exp(\mathbf{a}_{\text{vis}})\mathbf{v}_{\text{vis}}/D_2 \\
&+ \exp(\mathbf{a})\mathbf{v}_1/D_1 - \exp(\mathbf{a})\mathbf{v}_1/D_2\| \overset{(a)}{\approx} \|\exp(\mathbf{a}_{\text{vis}}(\mathbf{C}_{\text{vis}}^K)^\top)\mathbf{C}_{\text{vis}}^V \mathbf{v}_{\text{vis}} - \exp(\mathbf{a}_{\text{vis}})\mathbf{v}_{\text{vis}}\| \\
&\overset{(b)}{\leq} \|\exp(\mathbf{a}_{\text{vis}}(\mathbf{C}_{\text{vis}}^K)^\top)\mathbf{C}_{\text{vis}}^V \mathbf{v}_{\text{vis}} - \exp(\mathbf{a}_{\text{vis}})\mathbf{R}^\top \mathbf{R}\mathbf{v}_{\text{vis}}\| + \|\exp(\mathbf{a}_{\text{vis}})\mathbf{R}^\top \mathbf{R}\mathbf{v}_{\text{vis}} - \exp(\mathbf{a}_{\text{vis}})\mathbf{v}_{\text{vis}}\| \\
&\overset{(c)}{\leq} (1 + \epsilon)\|\mathbf{v}\| \|\exp\left(\mathbf{a}_{\text{vis}}\mathbf{C}_{\text{vis}}^K\right) - \exp(\mathbf{a}_{\text{vis}})\mathbf{R}^\top\| + \|\exp(\mathbf{a}_{\text{vis}})\mathbf{R}^\top \mathbf{R}\mathbf{v}_{\text{vis}} - \exp(\mathbf{a}_{\text{vis}})\mathbf{v}_{\text{vis}}\| \\
&\overset{(d)}{\leq} \|\exp(\mathbf{a}_{\text{vis}})\mathbf{R}^\top \mathbf{R}\mathbf{v}_{\text{vis}} - \exp(\mathbf{a}_{\text{vis}})\mathbf{v}_{\text{vis}}\| + o(\|\exp(\mathbf{a}_{\text{vis}})\|\|\mathbf{v}_{\text{vis}}\|) \\
&\overset{(e)}{\leq} \epsilon\|\exp(\mathbf{a}_{\text{vis}})\|\|\mathbf{v}_{\text{vis}}\| + o(\|\exp(\mathbf{a}_{\text{vis}})\|\|\mathbf{v}_{\text{vis}}\|)
\end{aligned}
\tag{29}
$$

The above, step (a) is based on the **Assumption 1**, *i.e.*, $\text{sum}(\exp(\mathbf{a}_1)) >> \text{sum}(\exp(\mathbf{a}_{\text{vis}}))$ and $\text{sum}(\exp(\mathbf{a}_1)) >> \text{sum}(\exp(\mathbf{a}_{\text{vis}}(\mathbf{C}_{\text{vis}}^K)^\top))$. The step (b) is based on the triangle inequality, and the step (c) leverages the Cauchy inequality and a version of JL Lemma from (Arriaga & Vempala, 2006). The step (d) utilizes the fact that exponential function is Lipschitz continuous in a compact region (Wang et al., 2020). The step (e) is based on Eq. 22. Applying the results in Eq. 29 to any row vector $\mathbf{a}$ of $\mathbf{A}$ and any column vector $\mathbf{v}$ of matrix $\mathbf{V}$, we can prove the Theorem 2.

# E    PROOF OF THEOREM 3

*Proof.* The main idea of the proof is based on the **Theorem 2** and the triangle inequality. Based on the definition of Eq. 9, the Pre-LLM compression strategy can be formulated as $\text{Com}(\mathbf{C}^Q, (\mathbf{C}^K), \mathbf{C}^V) = \text{softmax}\left(\mathbf{C}^Q \mathbf{A}(\mathbf{C}^K)^\top\right) \cdot \mathbf{C}^V \mathbf{V}$ and our ACM can be presented as $\text{Com}(\mathbf{I}, (\mathbf{C}^K)^\top, \mathbf{C}^V) = \text{softmax}\left(\mathbf{A}(\mathbf{C}^K)^\top\right) \cdot \mathbf{C}^V \mathbf{V}$. In the following, we use Pre-LLM and ACM to represent the detailed formulation for clarity. Then, for any $\mathbf{C}^Q$, $\mathbf{C}^K$, and $\mathbf{C}^V$, we have:

$$
\begin{aligned}
&\|\text{ACM} - \text{softmax}(\mathbf{A})\mathbf{V}\| \\
&\overset{(a)}{\leq} \|\text{ACM} - \text{Pre-LLM}\| + \|\text{Pre-LLM} - \text{softmax}(\mathbf{A})\mathbf{V}\| \\
&\overset{(b)}{<} \|\text{Pre-LLM} - \text{softmax}(\mathbf{A})\mathbf{V}\|
\end{aligned}
\tag{30}
$$

The step (a) is based on the the triangle inequality and the step (b) is based on the fact that $\|\text{softmax}\left(\mathbf{C}^Q \mathbf{A}(\mathbf{C}^K)^\top\right) \cdot \mathbf{C}^V \mathbf{V} - \text{softmax}\left(\mathbf{A}(\mathbf{C}^K)^\top\right) \cdot \mathbf{C}^V \mathbf{V}\| > 0$, since generally we have $\mathbf{C}^Q \mathbf{A}(\mathbf{C}^K)^\top \neq \mathbf{A}(\mathbf{C}^K)^\top$. The proof of Eq. 13 is similar to Eq. 30.

# F    MORE EXPERIMENTAL DETAILS AND RESULTS

## F.1    DETAILS OF EARLY COMPRESSION LIMITATION EXPERIMENTS

Here, we give more details about the experiment settings for Fig. 2. Specifically, we utilize CLIP-ViT-L/14-224px as the vision encoder and Vicuna-7B-v1.5 as the LLM. We adopt a two-stage training approach comprising a pre-training stage and an instruction supervised (SFT) fine-tuning stage, following the training parameters outlined in (Liu et al., 2023a). The number of slices is set to four, consistent with LLaVA-1.5-HD (Liu et al., 2023a). We test the trained model on four high-resolution benchmarks, *i.e.*, VQA-text(Singh et al., 2019), ChartQA val set (Masry et al., 2022), DocVQA val set (Mathew et al., 2021), InfoVQA val set (Mathew et al., 2022), and three general multimodal benchmarks including SEED (Li et al., 2023b), MMBench (Liu et al., 2023c), POPE (Li et al., 2023d). To explore the impact of dropping vision tokens at different layers within the LLM, we select a specific layer from the pre-trained model, discard 50% of the original vision tokens at that layer, and retain only these 50% in all subsequent layers. We sample a total of four layers at intervals from early to latter across all 32 layers of the LLM, specifically the 5th, 15th, 25th, and 30th layers. Additionally, we also include the 0th layer, which performs token dropping before the vision tokens are input into the LLM. We select three types of token-dropping ways to compare with the non-compression model. (i) average: dropping vision tokens based on averaged attention scores from all of 32 layers of the Vicuna-7B-v1.5; (ii) separate: dropping vision tokens based on averaged attention scores from the previous layer, and (iii) last: dropping vision tokens based on averaged attention scores from the last layer.

## F.2    STF DATASETS

Table 6 shows the detailed construction of the supervised instruction tuning dataset in Section 5.2. Our SFT data consists of four types: (i) caption data sampled from ShareGPT4V (Chen et al., 2023); (ii) Science data sampled from AI2D (Kembhavi et al., 2016) and ScienceQA (Lu et al., 2022); (iii) doc-related data sampled from ChartQA (Masry et al., 2022), DVQA  (Kafle et al., 2018), PlotQA (Methani et al., 2020), OCRVQA (Mishra et al., 2019), DocVQA (Mathew et al., 2021), InfoVQA (Mathew et al., 2022), synthdog-en (Kim et al., 2022) and TableFact (Chen et al., 2019); (iv) general data sampled from LLaVA (Liu et al., 2023a) and sharegpt4v (Chen et al., 2023).

## F.3    LAYER INDEX OF APPLYING ACM

In Section 5.4, we demonstrate that uniformly inserting ACM into different layers of the LLM yields the best performance. Here, we investigate the specific layer indexes of the early, middle, and latter layers within the LLM. We explore two different types: (i) continuous, *i.e.*, inserting ACMs into continuous layers of the LLM layers and (ii) interval, *i.e.*, inserting ACMs into the LLM layers with an interval. Results from Table 7 demonstrate that interval incorporation performs better in both 0.5B and 3B LLMs.

| Task | Dataset | # Sample |
|---|---|---|
| Captioning | ShareGPT4V (Chen et al., 2023) | 100K |
| Science | AI2D (Kembhavi et al., 2016) | 12K |
| | ScienceQA (Lu et al., 2022) | 12K |
| Doc QA | ChartQA (Masry et al., 2022) | 28K |
| | DVQA (Kafle et al., 2018), | 100K |
| | PlotQA (Methani et al., 2020) | 10K |
| | OCRVQA (Mishra et al., 2019) | 80K |
| | DocVQA (Mathew et al., 2021) | 49K |
| | InfoVQA (Mathew et al., 2022) | 14K |
| | synthdog-en (Kim et al., 2022) | 29K |
| | TableFact (Chen et al., 2019) | 14K |
| General QA | LLaVA (Liu et al., 2023a) | 150k |
| | sharegpt4v (Chen et al., 2023) | 665K |
| **Total** | - | **1.2M** |

Table 6: Summary of datasets for SFT in Section 5.2.

| LLM | early | middle | latter | average |
|---|---|---|---|---|
| Qwen2-0.5B | { 3, 5, 7, 9} | { 10-17 } | { 18, 20, 22 } | 61.08 |
| Qwen2-0.5B | { 3, 4, 8, 9} | { 10-17 } | { 20, 21, 22 } | 64.32 |
| Phi-3-3B | { 3, 4, 5, 11, 12, 13} | { 14-23 } | { 27, 28, 29, 30 } | 65.25 |
| Phi-3-3B | { 3, 5, 7, 9, 11, 13} | { 14-23 } | { 24, 26, 28, 30 } | 69.12 |

Table 7: Layer index of incorporating ACM. **average** means the average performance on high-resolution benchmarks. The rows in organe and blue represents applying ACM in the continuous and interval layers of LLM respectively.

### F.4 CASUAL MASK

In section 3.2, *i.e.*, Fig. 4 (b), we sampling the original casual mask based on the indexes of the selected vision tokens. Here, we also compare another implementation of the casual mask for the vision tokens, *i.e.*, setting high-resolution and low-resolution vision tokens to be non-causal, as shown in Fig. 8. Results from Table 8 show that using casual masks for both vision and text tokens can achieve better performance.

### F.5 TOKEN SELECTION STRATEGIES FROM ATTENTION WEIGHT

To assess token importance, we use the average attention score across all heads, as it provides a stable and comprehensive view of token importance by integrating multiple perspectives. In this section, we conduct the ablation studies for more different token selection strategies.

#### F.5.1 MULTI-HEADS

In this section, we ablate the effect of attention scores from muli-heads: (i) **Specific heads**: Randomly selecting one head. (ii) **Separate**: Performing token selection independently within each head. As shown in Table 9, **Average (Ours)** achieves the highest and most consistent performance. Averaging

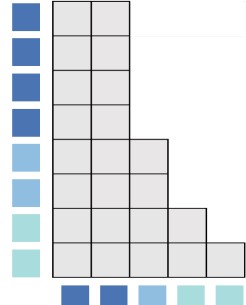

Figure 8: **Non-Casual for high-resolution and low-resolution vision tokens.**

| Model | General | | | High-Resolution | | | |
|---|---|---|---|---|---|---|---|
| | SEED | POPE | MME | VQA-text | ChartQA | DocVQA | InfoVQA |
| Casual | **63.5** | **87.6** | **1480.3** | **58.5** | **46.1** | **45.2** | **31.6** |
| Non-Casual | 62.7 | 86.5 | 1455.2 | 57.3 | 45.4 | 44.5 | 29.5 |

Table 8: Comparison with the casual or non-casual mask for vision tokens

| way | general | high-resolution |
|---|---|---|
| Average (Ours) | **75.04** | **45.35** |
| Specific-1 | 69.53 | 34.29 |
| Specific-2 | 71.43 | 38.21 |
| Specific-3 | 70.21 | 37.55 |
| Separate | 73.51 | 44.05 |

Table 9: Comparison of token selection strategies across different attention aggregation methods. Specific-1', 'Specific-2', and 'Specific-3' represent three specific heads randomly selected in our experiments.

provides a comprehensive view, combining insights from multiple heads to capture both global and local token importance. **Specific** shows variable performance across different heads, as each head may focus on unique aspects of the data. Selecting heads suitable for all tasks proves challenging due to this variability. **Separate** performs better than Specific but still falls short of our method. Since each head typically captures local information, analyzing them separately limits the ability to assess global token importance.

| Method | general | high-resolution |
|---|---|---|
| Vision-to-Text (Ours) | **75.04** | **45.35** |
| Vision-to-Vision | 72.47 | 42.86 |
| Text-to-Vision | Not Applied | |

Table 10: Performance comparison between vision-to-text and vision-to-vision token selection strategies.

### F.5.2 DIFFERENT TYPES OF TOKENS

In this section, we analyze the **attention weight distributions** across token types: **vision-to-vision**, **vision-to-text**, and **text-to-vision**, providing insights into the token compression mechanism. From Fig. 2 (b), we can have the follow observations: (i) **Vision-to-Vision**: Dense attention patterns focus on local visual relationships but lack the ability to capture multimodal dependencies. (ii) **Vision-to-Text (Ours)**: Selectively attends to text-relevant vision tokens, effectively integrating multimodal cues and enhancing task performance. (iii) **Text-to-Vision**: Current MLLMs concatenate vision and text tokens in a fixed order, making direct analysis of text-to-vision distributions challenging. As shown in Table 10, Vision-to-Text (Ours) Outperforms vision-to-vision attention weights, demonstrating superior performance on both general and high-resolution benchmarks. The reason may due to that Vision-to-Vision relies solely on single-modal visual information, which limits its effectiveness in capturing multimodal dependencies required for complex tasks.

## F.6 TOKEN SELECTION WITH ENTROPY

To strengthen our analysis, we incorporated entropy measurements to quantitatively support our token selection strategy. **Entropy Computation:**

Entropy measures the uncertainty of attention weights and is calculated as:

$$H = -\sum_{i=1}^{n} a_i \log(a_i),$$

where $a_i$ is the normalized attention weight for token $i$, and $n$ is the total number of tokens. High-entropy tokens indicate diverse information, while low-entropy tokens reflect concentrated, less complex relationships.

**Experiment Design:** We tested two settings: (i) **High-entropy:** Selecting top-K tokens with the highest entropy values. (ii) **Low-entropy:** Selecting top-K tokens with the lowest entropy values. We also computed the **overlap percentage** between entropy-based selections and our method to evaluate alignment.

From Table 11, we can observe that: (i) **High-entropy tokens outperform low-entropy tokens**, showing they capture richer features. (ii) **Performance correlates with overlap**: High-entropy tokens (71.2% overlap) achieve comparable results to ours, while low-entropy tokens (40.1% overlap) significantly underperform. (iii) **Compared to Our Method**: Our method outperforms high-entropy selection, suggesting its token selection balances diversity (high entropy) and text-relevant information. These results confirm that our strategy inherently selects high-information-content tokens as well as the text-guided information.

| way | *overlap (%)* | *general* | *high-resolution* |
|---|---|---|---|
| Ours | 100.0 | **75.04** | **45.35** |
| High-Entropy | 71.2 | 73.06 | 42.33 |
| Low-Entropy | 40.1 | 71.12 | 38.21 |

Table 11: Performance comparison between our method and entropy-based token selection strategies. High-entropy tokens show closer alignment with our method and achieve superior results compared to low-entropy tokens.

## F.7 COMPARISON RESULTS ON V* BENCH

To further validate the effectiveness of our method, we also report the comparison results in Table 12. **All results are tested based on the models used in Table 2.** Results demonstrate that our method still outperforms the existing approaches, indicating that our method can work for various domains.

| Model | Attribute | Spatial | Overall |
|---|---|---|---|
| Full | 46.8 | 63.2 | 53.1 |
| FlexAttention | 43.8 | 60.5 | 49.6 |
| Avg-Pool | 43.9 | 60.6 | 50.5 |
| C-Abstractor | 43.5 | 60.9 | 50.2 |
| Ours | **45.7** | **62.8** | **52.2** |

Table 12: Comparison with different methods on V* bench.

## F.8 THE NUMBER OF COMPRESSION RATIOS

In this section, we analyze the number of compression ratios, *i.e.*, $\{r_i, r_j, r_p\}$, which determine how much vision tokens are dropped. Table 13 demonstrates the performance and efficiency trade-offs across different compression ratios. Excessive compression, such as $\{2, 8, 16\}$, significantly degrades

performance in high-resolution tasks despite marginally improving efficiency. Therefore, we selected $\{2, 4, 8\}$ as the optimal setting, providing a balance between efficiency and robust performance across general and high-resolution tasks.

| $\{r_i, r_j, r_p\}$ | Time (ms) | general | high-resolution |
|---|---|---|---|
| $\{1, 2, 4\}$ | 532 | 75.52 | 45.86 |
| $\{2, 4, 8\}$ | 515 | 75.04 | 45.35 |
| $\{2, 8, 16\}$ | 508 | 74.32 | 43.08 |

Table 13: Performance and efficiency with different compression ratios.

### F.9 COMPARISON WITH PREVIOUS AND CURRENT LAYERS

In our approach, we use the attention map from the previous non-compression layer to guide vision token compression, primarily to reduce computational and memory overhead. Using the current layer's attention map would require performing full attention between the query and key tokens before compression, significantly increasing resource usage. As shown in Table 14, we compare the performance and time efficiency of using the previous layer's attention map (ours) versus the current layer's. Both methods achieve nearly identical performance, but the current layer incurs additional computation time.

| Method | Time (ms) | general | high-resolution |
|---|---|---|---|
| Previous Layer (Ours) | 515 | 75.04 | 45.35 |
| Current Layer | 524 | 75.05 | 45.39 |

Table 14: Performance and efficiency comparison between token selection using the previous layer's attention map (ours) and the current layer's attention map.

### F.10 TRAINING TIME

In Table 2, we report the single-forward pass time as an indicator of both training and inference efficiency. In Table 15, we have now included total training time comparisons across models, demonstrating that our model achieves approximately 82.9% of the training time required by the full model.

| Method | Training time | Inference time | Avg performance |
|---|---|---|---|
| Full | 32.2 h | 621 ms | 48.0 |
| Q-former | 25.1 h | 507 ms | 30.6 |
| Avg-pooling | 24.7 h | 461 ms | 39.1 |
| FlexAttn | 25.4 h | 505 ms | 32.5 |
| LLaVA-UHD | 24.9 h | 470 ms | 36.0 |
| C-Abstractor | 25.0 h | 492 ms | 27.6 |
| FastV | 25.2 h | 499 ms | 39.9 |
| Ours | 26.7 h | 512 ms | 45.4 |

Table 15: Comparison with the efficiency and performance. All models are trained in one epoch on 16 V100 GPUs.

### F.11 TOKEN SELECTION BASED ON VISUAL INFORMATION

We leverage the text-guided token selection for two reasons: (i) In the casual LLMs, the last text token receives holistic information from the whole previous tokens; (ii) Text-guided selection can efficiently filter the instruct-related tokens from noisy tokens.

In this section, to verify the potential benefits of incorporating visual information, we conducted additional experiments with ToMe [1] and Dynamic ViT [2], which explicitly consider visual complexity. Specifically:

- **ToMe[1]**: A similarity-based token merging method that preserves visual structure.
- **Dynamic ViT[2]**: A dynamic token pruning approach guided by visual importance.
- **ToMe+Ours**: We combined ToMe with our method by using our text-guided approach to select key tokens and ToMe to merge remaining tokens.

From Table 16, we have the following observations: (i) ToMe performs slightly better (+0.11%) on low-resolution tasks but is slower due to additional merging steps. (ii) Ours excels on high-resolution tasks (+1.42%) by efficiently filtering relevant tokens and reducing noise. (iii) Combining ToMe with our method improves performance slightly but adds significant computational cost.

In conclusion, while visual complexity-based methods like ToMe show certain advantages in specific settings, our approach strikes a better balance between performance and efficiency, particularly for high-resolution tasks. The additional experiments validate that text-guided selection remains an effective and practical choice for diverse tasks.

| way | Times (ms) | general | high-resolution |
|---|---|---|---|
| Attention-weight (ours) | 515 | 75.04 | **45.35** |
| ToMe | 564 | **75.15** | 43.93 |
| Dynamic ViT | 520 | 73.51 | 42.82 |
| Ours + ToMe | 571 | 75.21 | 45.44 |

Table 16: Comparison with different compression ways that use visual nuances in our ACT-In-MLLM.

### F.12 COMPARISON WITH HIGHER COMPRESSION RATIO

Table 17 reports additional experiments with both FastV and our method to achieve approximately 60% compression, aiming for similar time efficiency. The results indicate that our method outperforms FastV by 3.43% under comparable efficiency conditions.

| Model | Times (s) | Memory (GB) | high-resolution |
|---|---|---|---|
| Full | 1.95 (100.0%) | 26.8G (100.0%) | 52.12 |
| FastV [1] | 1.21 (61.8%) | 22.5G (83.9%) | 44.82 |
| Ours | 1.27 (62.6%) | 23.0G (86.0%) | 48.25 (+3.43) |

Table 17: Comparison of efficiency and performance between FastV and our method.

### F.13 COMPARISON WITH FASTV AND OURS UNDER MORE VISION TOKENS

In Table 2, we deliberately used 512 tokens, as we believe thatretaining fewer vision tokens provides a better comparison of how well different compression methods preserve critical information. Additionally, as shown in Fig. 6, increasing the number of retained tokens improves performance consistently, further highlighting the effect of the number of reserved vision tokens for the performance, *i.e.*, retaining more vision tokens leads to higher performance.

We also conduct new experiments on LLaVA-NeXT-7B (Liu et al., 2024b) with both FastV and our method retaining 1440 tokens. The results show that our method outperforms FastV by 2.2%

| Model | Efficiency | | General | | | High-Resolution | | | | |
|---|---|---|---|---|---|---|---|---|---|---|
| | Times(ms) | Memory(GB) | SEED | POPE | MME | VQA-text | ChartQA | DocVQA | InfoVQA | Average |
| Open-LLava-Next | 926(**100.0%**) | 24.3(**100.0%**) | 70.3 | 85.8 | 1533.5 | 67.1 | 64.2 | 70.0 | 34.5 | 58.9 |
| FastV w/o train | 579 (**62.5%**) | 21.5(**88.4%**) | 69.6 | 85.5 | 1502.0 | 67.0 | 60.5 | 62.1 | 33.1 | 55.7 |
| Ours w/o train | 592 (**63.9%**) | 22.1(**90.9%**) | 70.1 | 86.1 | 1532.2 | 67.3 | 63.1 | 66.9 | 33.4 | 57.9 |

Table 18: Comparison with applying Fastv and ours as the inference-only strategy based on Open-LLaVA-Next.

on high-resolution tasks while maintaining comparable inference times. These results confirm our method's superior ability to preserve information under different vision token settings.

## F.14 COMPARISON WITH FASTV AND OURS BASED ON LLAVA-1.5 TRAINING DATA

We conduct additional experiments using the original training data from LLaVA-1.5-7B (Liu et al., 2023a), applying both our method and FastV during training and inference. The results demonstrate that our method consistently outperforms FastV and achieves performance comparable to Full on both general and high-resolution benchmarks.

Notably, since LLaVA-1.5 lacks high-resolution training data, the performance gap between our method and FastV is smaller compared to experiments with augmented high-resolution data. For Open-LLaVA-NeXT[2] (since the training data from LLaVA-Next (Liu et al., 2024b) is not released), when trained on high-resolution data, our method exhibits a larger performance advantage over FastV, highlighting its superior ability to leverage additional information.

| Model | Training/Inference Times | General | | | High-Resolution | | | | |
|---|---|---|---|---|---|---|---|---|---|
| | | SEED | POPE | MME | VQA-text | ChartQA | DocVQA | InfoVQA | Average |
| LLava-1.5 | 334ms/20.5h | 66.1 | 85.9 | 1510.7 | 58.2 | 18.2 | 21.2 | 20.6 | 29.6 |
| FastV | 273ms/17.4h | 65.2 | 85.2 | 1470.5 | 57.1 | 17.2 | 19.1 | 19.3 | 28.2 |
| Ours | 286ms/18.1h | 66.3 | 85.8 | 1510.2 | 58.0 | 18.3 | 21.0 | 20.2 | 29.4 |
| Open-LLava-Next | 926ms/63.2h | 70.9 | 86.2 | 1535.4 | 67.3 | 64.6 | 69.5 | 33.4 | 58.7 |
| FastV | 579ms/45.8h | 69.8 | 86.3 | 1489.6 | 66.5 | 60.1 | 62.8 | 32.8 | 55.6 |
| Ours | 592ms/48.5h | 70.2 | 86.5 | 1530.4 | 67.4 | 63.8 | 67.5 | 33.2 | 58.0 |

Table 19: Comparison between FastV and our method on LLaVA-1.5 and Open-LLaVA-Next during both training and inference.

## F.15 VISUALIZATION

In Fig. 9, Fig. 10 and Fig. 11, we show visualization results to compare our method with existing vision compression approaches, *i.e.*, average-pooling from (Liu et al., 2024b) and FlexAttention (Li et al., 2024b).

---

[2]https://github.com/xiaoachen98/Open-LLaVA-NeXT

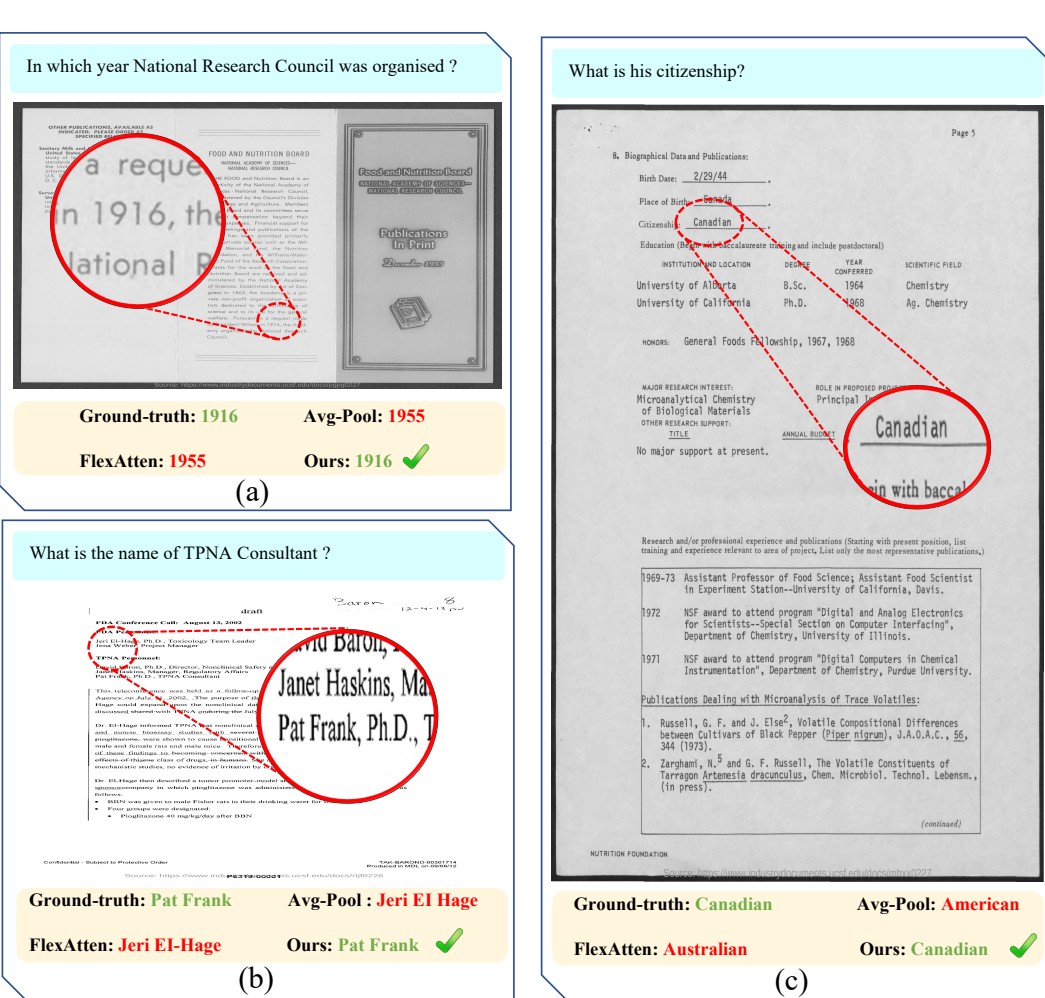

Figure 9: **Qualitative results from DocVQA (Mathew et al., 2021).** We compare ACT-IN-LLM with Average-pooling from (Liu et al., 2024b) and FlexAttn (Li et al., 2024b)

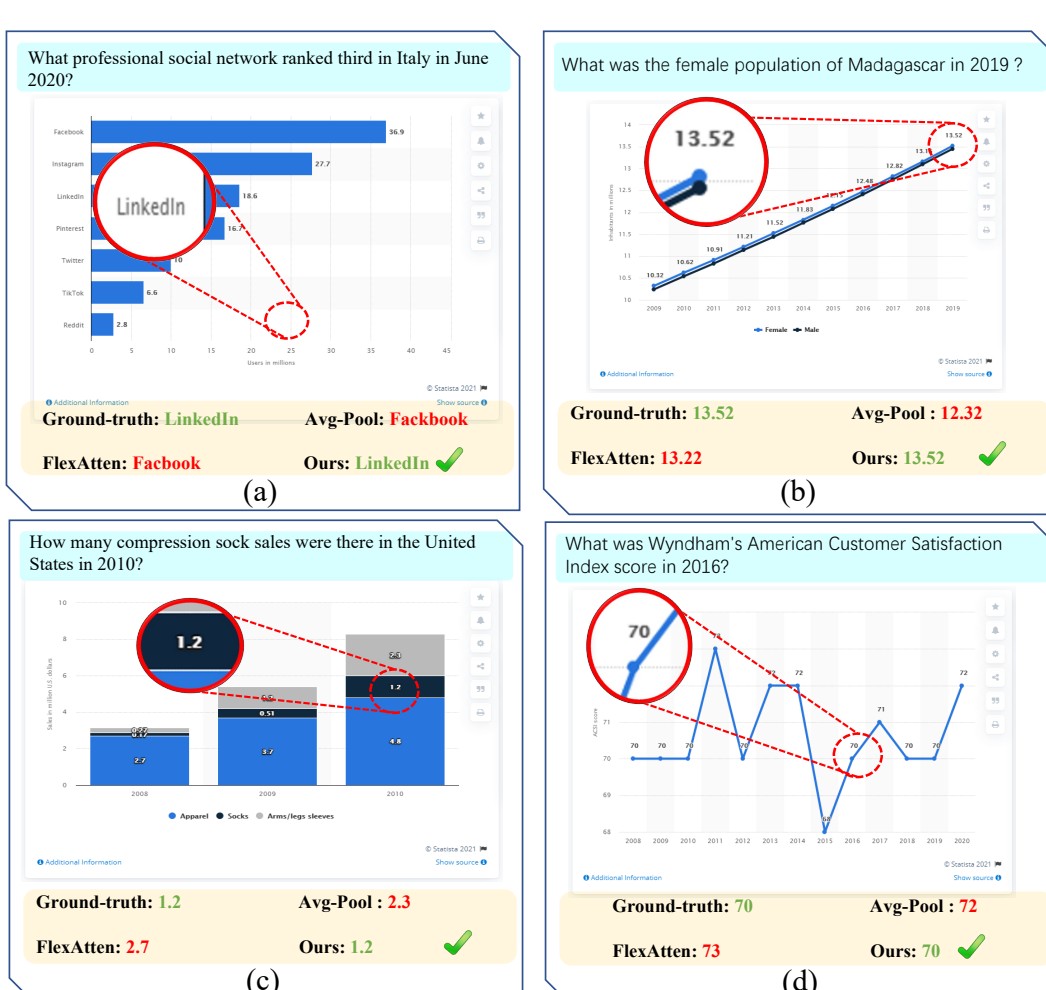

Figure 10: **Qualitative results from ChartQA (Masry et al., 2022).** We compare ACT-IN-LLMwith Average-pooling from (Liu et al., 2024b) and FlexAttn (Li et al., 2024b)

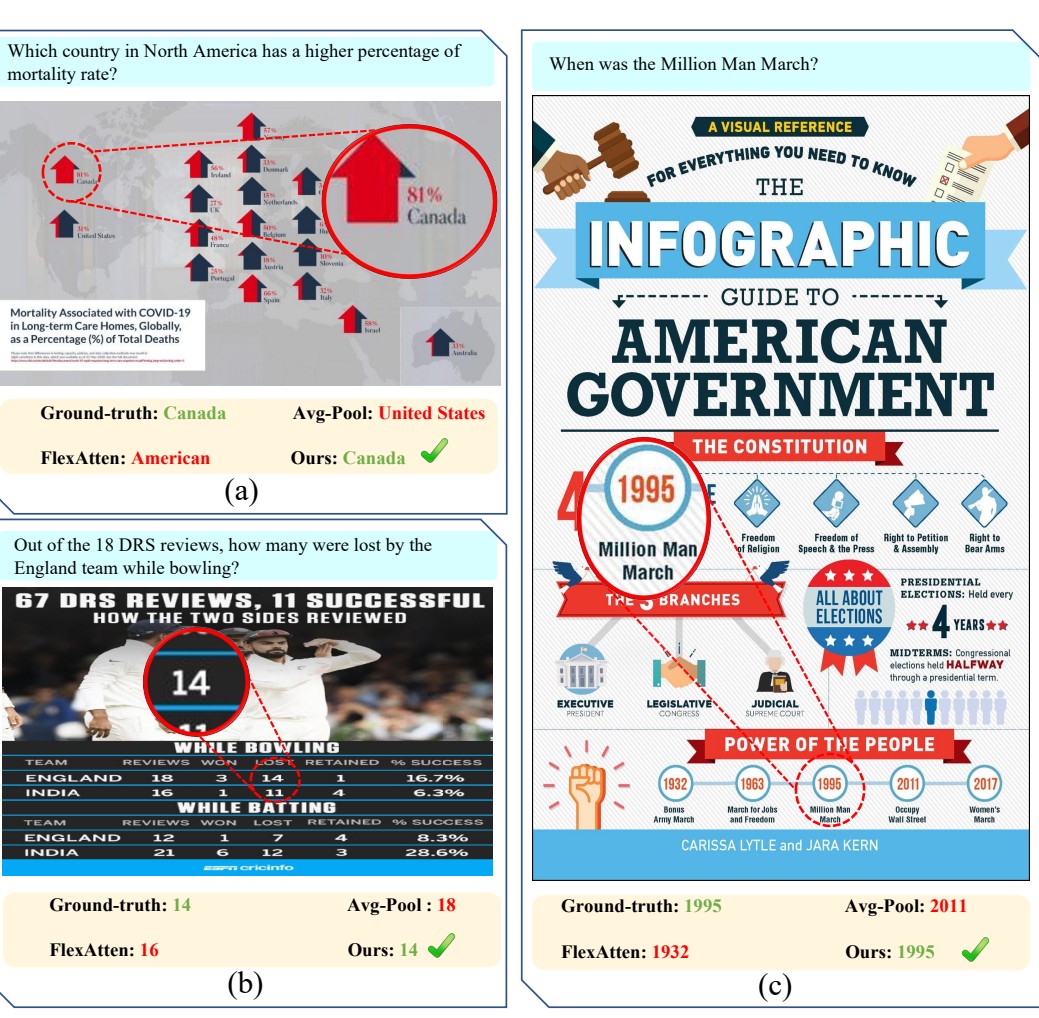

Figure 11: **Qualitative results from InfoVQA (Mathew et al., 2022).** We compare ACT-IN-LLM with Average-pooling from (Liu et al., 2024b) and FlexAttn (Li et al., 2024b)

