# OpenReview forum: "ACT-IN-LLM: Adaptively Compression Vision Tokens in LLM for High-Resolution Multimodal Large Language Models"
_ICLR.cc/2025/Conference — Submitted to ICLR 2025_

### Official Review · Reviewer_9yBs · 2024-10-30

**Soundness:** 3
**Presentation:** 3
**Contribution:** 2
**Rating:** 3
**Confidence:** 4

**Summary:**

The paper examines the limitations of token compression strategies in MLLMs when processing high-resolution visual inputs. It presents ACT-IN-LLM, an approach designed to address these limitations by adaptively compressing visual tokens across LLM layers, contrasting with existing methods that apply compression before token input to the LLM. The authors claim this layer-wise, interaction-guided compression effectively preserves essential visual information, improving model accuracy while reducing computational load. Experiments indicate significant performance gains over prior compression strategies and competitive results with non-compression methods, highlighting ACT-IN-LLM’s potential to enhance high-resolution MLLM efficiency and scalability.

**Strengths:**

By constructing a unified formulation of compression methods and analyzing low-rank approximations, the paper provides a robust theoretical framework supporting its method. Experimental results convincingly demonstrate that ACT-IN-LLM outperforms existing pre-LLM compression and interaction-based methods, achieving competitive accuracy while significantly reducing token count and computational costs. Notably, the proposed method shows good scalability, with consistent gains observed across larger model sizes and datasets. These strengths suggest that ACT-IN-LLM offers a practical, efficient solution for high-resolution MLLM applications, and the work is both well-structured and empirically solid.

**Weaknesses:**

Relying solely on text-guided token selection may limit the model's adaptability, as it could overlook the inherent complexity of visual information itself. Without considering factors like scene detail or object density, the compression strategy might miss important visual nuances, potentially affecting performance across diverse tasks.

**Questions:**

Thanks for the authors' valuable exploration in this area. I have several concerns, and if these can be addressed, I would like to raise my rating.

1. The reported compression rate seems relatively low (the proposed method achieves 83% of the full model's performance and 94% of its memory usage according to Table 2). Would it be possible for the authors to provide results with a higher compression ratio (around 60%, for example) to more effectively demonstrate the advantages of the proposed method?
2. It appears that FastV[1] is not included in the comparisons. Could the authors consider providing comparisons with FastV, particularly at higher compression ratios (such as around 60%)?
3. The current reliance on text-guided token selection may have limitations. Have the authors considered incorporating the complexity of visual information into the compression strategy?

[1] https://arxiv.org/pdf/2403.06764

---

### Official Review · Reviewer_xyS7 · 2024-10-31

**Soundness:** 3
**Presentation:** 3
**Contribution:** 2
**Rating:** 3
**Confidence:** 4

**Summary:**

The proposed ACT-IN-LLM method improves Multimodal Large Language Models (MLLMs) by adaptively compressing vision tokens within different layers, unlike traditional methods that reduce tokens before reaching the LLM. This approach preserves all tokens throughout layers, selectively compressing only key and value tokens in self-attention to maintain critical information while reducing computational load.

**Strengths:**

1. This paper provides a valuable perspective on why early token deletion should be avoided.
2. The writing is generally smooth, and the presentation of figures and tables is visually appealing, contributing positively to the overall presentation of the paper.
3. The motivation section reads coherently and introduces the problem that needs to be addressed in a natural manner.

**Weaknesses:**

1. The paper dedicates significant space to comparisons with traditional methods that compress tokens before using LLMs. However, it overlooks an essential baseline, FastV [1], which also compresses image tokens within the LLM itself and allows for direct comparison of training results. This omission makes the paper less convincing.
2. The paper claims "reducing training/inference time," but does not provide any data demonstrating training time reduction.
3. The proposed strategy appears usable without training; therefore, it would be beneficial to include results showing inference acceleration without additional training.
4. The existence of Table 3 is quite awkward: first, there are numerous gaps in the table, and second, the training data is entirely different, making these models incomparable.
[1]An Image is Worth 1/2 Tokens After Layer 2: Plug-and-Play Inference Acceleration for Large Vision-Language Models, https://arxiv.org/abs/2403.06764

**Questions:**

1. Based on the results presented, all query tokens are retained, while key and value tokens are compressed in the self-attention mechanism. If the value tokens are compressed, the number of tokens outputted from the attention block should match the number of compressed value tokens. Then why are all image tokens still preserved when entering the final LM head? Please explain this in detail.
2. Please provide a detailed explanation of how your strategy reduces computational load across various components.

---

> ### Public Comment · ~Tao_Zhang27 · 2025-10-16
> **Confusions regarding Q1**
>
> “If the value tokens are compressed, the number of tokens outputted from the attention block should match the number of compressed value tokens. ”
> The argument contains a fundamental misunderstanding of the self-attention mechanism in Transformers. In standard attention, the output sequence length is determined solely by the number of queries, not by the number of keys or values.

---

### Official Review · Reviewer_k8Dx · 2024-11-04

**Soundness:** 3
**Presentation:** 4
**Contribution:** 3
**Rating:** 8
**Confidence:** 5

**Summary:**

This paper introduces a novel Adaptive Compression Module (ACM) designed to dynamically reduce the number of high-resolution image tokens for key/value during the forward pass. The ACM leverages attention maps to identify and retain the most relevant high-resolution tokens, preserving only the top k tokens for key/value. Experimental results demonstrate the efficiency and effectiveness of this approach.

**Strengths:**

1. The paper is well-written and easy to follow, offering a clear mathematical proof to demonstrate the theoretical effectiveness of the proposed method.

2. The proposed approach is both intuitively and mathematically sound.

3. The experimental results are thorough, complemented by a comprehensive ablation study.

**Weaknesses:**

The evaluation benchmark is somewhat limited. Adding more benchmarks, such as visual grounding benchmarks and other non-text-related high-resolution benchmarks like V* Bench, would facilitate more comprehensive evaluations.

**Questions:**

1. As you are progressively shrinking the ratio $r_i$, $r_j$ and $r_p$, how do you determine where and how much should you shrink?

2. Why do you use the attention map from the previous layer to guide vision token compression instead of the current layer's attention map?

---

### Official Review · Reviewer_QYE1 · 2024-11-04

**Soundness:** 3
**Presentation:** 2
**Contribution:** 3
**Rating:** 8
**Confidence:** 4

**Summary:**

This paper addresses the challenge of processing high-resolution images in multimodal large language models (MLLMs) by introducing ACT-IN-LLM, a novel adaptive compression strategy for vision tokens. Unlike existing pre-LLM compression methods that reduce tokens before LLM processing, ACT-IN-LLM performs compression within different LLM layers through an adaptive compression module (ACM). The method selectively compresses key and value tokens in the self-attention mechanism while retaining all tokens across layers, guided by each layer's final token that encodes the complete multimodal context. The authors provide theoretical analysis demonstrating that their key-value compression approach achieves better low-rank approximation compared to existing compression techniques. Experimental results across various LLM sizes (0.5B to 7B parameters) show that ACT-IN-LLM achieves a 6.2% improvement over existing compression methods while maintaining competitive performance with non-compression models, reducing training/inference time by approximately 20% and vision tokens by 60%.

**Strengths:**

1. Originality:
- The paper introduces a novel in-layer compression approach, departing from conventional pre-LLM compression methods. This represents a fundamental shift in how vision tokens are handled in MLLMs.
- The adaptive compression strategy that operates within different LLM layers is an innovative solution to the high-resolution image processing challenge.

2. Technical Quality:
- The work provides solid theoretical foundations by analyzing token compression through the lens of low-rank approximation in self-attention mechanisms.
- The authors conduct detailed empirical studies to demonstrate why early-layer compression is suboptimal, supporting their approach with concrete evidence.
- The technical approach is well-motivated through empirical observations about token importance varying across layers.

3. Solid Experiments:
- The method achieves substantial practical improvements, reducing training/inference time by 20% and vision tokens by 60% while maintaining competitive performance.
- The 6.2% performance improvement over existing compression techniques represents a significant advancement in high-resolution image processing for MLLMs.
- The experimental validation is comprehensive, spanning multiple model sizes (0.5B to 7B parameters) and various benchmarks.

**Weaknesses:**

1. Methodological Clarity and Analysis:
- The paper lacks clear explanation of how attention weights across multiple heads are handled in their analysis. This is crucial for understanding their token importance assessment methodology, as different aggregation methods (averaging across heads, selecting specific heads, or analyzing heads separately) could lead to different conclusions about token importance.
- The analysis of attention weight distributions between different types of tokens (vision-to-vision, vision-to-text, text-to-vision) is missing, which could provide deeper insights into the token compression mechanism.
- The authors could strengthen their analysis by including entropy measurements of attention weights for different tokens, which would provide quantitative support for their token selection strategy.

2. Technical Presentation:
- The paper introduces concepts like "high-resolution" and "low-resolution" tokens without first establishing the context of LLaVA's AnyRes visual encoding scheme. This may create confusion for readers not familiar with the underlying visual encoding mechanisms in multimodal LLMs.

**Questions:**

See "Weakness" 1.

---

### Meta-Review · Area_Chair_GLR9 · 2024-12-24

**Metareview:**

The paper proposes ACT-IN-LLM, an adaptive compression method for vision tokens in multimodal large language models, aiming to balance computational efficiency and high-resolution performance. Reviewers recognized the paper's originality, robust theoretical grounding, and extensive empirical validation. Strengths included the novel in-layer compression strategy and the comprehensive analysis supporting its efficacy. However, concerns persisted about the lack of flash-attention compatibility, inconsistent baseline comparisons, and issues with reproducibility due to reliance on augmented datasets.

The authors made significant efforts during the rebuttal, adding new experiments, addressing entropy-based token selection, and expanding comparisons with FastV under consistent settings. They clarified many methodological aspects and provided additional insights into training and inference efficiency. Despite these efforts, some critical concerns from reviewers, such as flash-attention limitations and inconsistent experimental baselines (notably in Table 3), remained partially unresolved. Reviewer xyS7 and 9yBs, in particular, emphasized the importance of these issues for practical scalability and scientific rigor. While the authors defended their work convincingly, the concerns about efficiency and uncontrolled comparisons were not fully mitigated.

Considering the balance between innovation, the thoroughness of the rebuttal, and unresolved issues, the AC recommends rejection at this point. The paper offers meaningful contributions and could be considered for acceptance in future venues. In its current form, however, more efforts need to be taken to address the remaining concerns and fully integrate all the valuable feedback.

**Additional Comments On Reviewer Discussion:**

Please refer to the meta review.

---

### Decision · Program_Chairs · 2025-01-22

Reject